# Physical Gold Nanoparticle-Decorated Polyethylene Glycol-Hydroxyapatite Composites Guide Osteogenesis and Angiogenesis of Mesenchymal Stem Cells

**DOI:** 10.3390/biomedicines9111632

**Published:** 2021-11-06

**Authors:** Chiung-Chyi Shen, Shan-hui Hsu, Kai-Bo Chang, Chun-An Yeh, Hsiang-Chun Chang, Cheng-Ming Tang, Yi-Chin Yang, Hsien-Hsu Hsieh, Huey-Shan Hung

**Affiliations:** 1Department of Neurosurgery, Neurological Institute, Taichung Veterans General Hospital, Taichung 407204, Taiwan; shengeorge@yahoo.com (C.-C.S.); jean1007@gmail.com (Y.-C.Y.); 2Department of Physical Therapy, Hung Kuang University, Taichung 433304, Taiwan; 3Basic Medical Education Center, Central Taiwan University of Science and Technology, Taichung 406053, Taiwan; 4Institute of Polymer Science and Engineering, National Taiwan University, Taipei 10617, Taiwan; shhsu@ntu.edu.tw; 5Graduate Institute of Biomedical Science, China Medical University, Taichung 40402, Taiwan; kbwork2021@gmail.com (K.-B.C.); fireleafmaple@hotmail.com (C.-A.Y.); hs0603@gmail.com (H.-C.C.); 6College of Oral Medicine, Chung Shan Medical University, Taichung 40201, Taiwan; ranger@csmu.edu.tw; 7Blood Bank, Taichung Veterans General Hospital, Taichung 407204, Taiwan; hhhsu@vghtc.gov.tw; 8Translational Medicine Research, China Medical University Hospital, Taichung 40402, Taiwan

**Keywords:** polyethylene glycol, hydroxyapatite, physical gold nanoparticle, mesenchymal stem cells

## Abstract

In this study, polyethylene glycol (PEG) with hydroxyapatite (HA), with the incorporation of physical gold nanoparticles (AuNPs), was created and equipped through a surface coating technique in order to form PEG-HA-AuNP nanocomposites. The surface morphology and chemical composition were characterized using scanning electron microscopy (SEM), atomic force microscopy (AFM), UV–Vis spectroscopy (UV–Vis), Fourier transform infrared spectroscopy (FTIR), X-ray photoelectron spectroscopy (XPS), and contact angle assessment. The effects of PEG-HA-AuNP nanocomposites on the biocompatibility and biological activity of MC3T3-E1 osteoblast cells, endothelial cells (EC), macrophages (RAW 264.7), and human mesenchymal stem cells (MSCs), as well as the guiding of osteogenic differentiation, were estimated through the use of an in vitro assay. Moreover, the anti-inflammatory, biocompatibility, and endothelialization capacities were further assessed through in vivo evaluation. The PEG-HA-AuNP nanocomposites showed superior biological properties and biocompatibility capacity for cell behavior in both MC3T3-E1 cells and MSCs. These biological events surrounding the cells could be associated with the activation of adhesion, proliferation, migration, and differentiation processes on the PEG-HA-AuNP nanocomposites. Indeed, the induction of the osteogenic differentiation of MSCs by PEG-HA-AuNP nanocomposites and enhanced mineralization activity were also evidenced in this study. Moreover, from the in vivo assay, we further found that PEG-HA-AuNP nanocomposites not only facilitate the anti-immune response, as well as reducing CD86 expression, but also facilitate the endothelialization ability, as well as promoting CD31 expression, when implanted into rats subcutaneously for a period of 1 month. The current research illustrates the potential of PEG-HA-AuNP nanocomposites when used in combination with MSCs for the regeneration of bone tissue, with their nanotopography being employed as an applicable surface modification approach for the fabrication of biomaterials.

## 1. Introduction

According to recent studies, poly (lactic-*co*-glycolic acid) (PLGA)-based artificial bone-substitute materials are highly attractive due to their better biocompatibility, degradability, and mechanical properties which enhance bone regeneration [1,2]. A hydrogel-based drug delivery system combined with nanoparticles has been used as a drug reservoir in order to improve the treatment effect [3]. Stem cells have been differentiated into chondrocytes with the assistance of either chondrogenic factors or scaffolds [4]. Moreover, a previous study indicated that scaffolds and growth factors are two important components that influence bone regeneration. PLGA/HA composite scaffolds with controlled gene release and Dox-regulated gene expression by electrical stimulation promoted cell proliferation and osteogenic differentiation in vitro while enhancing bone repair in an in vivo rabbit radial defect model [5].

A perfect biomaterial for bone substitution should be biocompatible, biodegradable, and osteoconductive [6]. Currently, different types of bone biomaterials including ceramics, polymers, decalcified matrices, bioactive glasses, hydroxyapatite (HA), and collagen (Col) have been widely applied for use as essential components for bone tissue regeneration [7,8,9,10]. Additionally, certain principal polymers, such as polyesters, poly (L-lactic acid) (PLA), poly (glycolide) (PGA), and poly (lactide-co-glycolide) (PLGA), have also been extensively certified due to their superior mechanical and physical properties for bone tissue regeneration [11,12]. However, the lack of sufficient biological activity in these polymers implies they will still encounter limitations. For example, HA ceramics have a disadvantage due to their poor hydrophilic properties for bone tissue regeneration [13]. For this reason, integration involving the advantages of bone biomaterials offers a new strategy towards ameliorating biomimic functions and mechanical properties and remains a critical issue for now.

Hydroxyapatite is the main mineral component in bone tissue. HA has a distinct surface effect, meaning it is prevalent in a wide range of biological molecules, such as proteins or growth factors [14,15]. Therefore, in order to achieve the application of the HA polymer, appropriate interfacial properties between the polymer and mineral components are required in order to accomplish mechanical, chemical, and biological capabilities [16,17]. In addition, the inherent property of HA that encourages osteoblast or mesenchymal stem cell recruitment and growth leads to osteogenesis [18,19]. Recently, the HA polymer has been widely applied as a replacement bone material, including HA-PLA [20], collagen [21], polyamides [22], polyα-caprolactone [23], and poly (α-hydroxy acid) [24]. A report indicated that HA-PLGA nanocomposites are created by gas formation and particle leaching methods with 3D fiber deposition, which can promote bone regeneration [6]. Another study also demonstrated that crosslinked poly (ε-caprolactone) (PCL) administers iron into hydroxyapatite (Fe-HA) nanoparticles to form 3D PCL/Fe-HA scaffolds, which can prompt the mechanical and mass transfer of properties for bone tissue repair, as evaluated in a rabbit model [25]. However, the remaining 3D PCL/Fe-HA scaffolds encountered lower uniformity and poor mechanical properties. Additionally, chitosan-coated iron oxide nanoparticles enhanced osteoblast proliferation, decreased cell membrane damage, and promoted cell differentiation. These works indicate that the presence of organic/inorganic nanoparticles can promote osteogenic differentiation [26]. The incorporation of HA nanoparticles into a poly(ethylene glycol) (PEG)-based hydrogel to guide its mineralization potentiality has been well elucidated for bone tissue regeneration [27]. A previous study demonstrated the effects of cellulose nanofibers (CNFs) and hydroxyapatite (HA) nanoparticles incorporated into a poly(ε-caprolactone) (PCL) matrix. The surface wettability and mechanical properties of PCL scaffolds were improved by CNFs, and further with HA addition. The results of the compressive and elastic moduli also verified the excellent properties for bone tissue engineering [28]. Furthermore, the addition of HA and CNFs did not weaken the biocompatibility of PCL/CNF/HA nanocomposites [28]. Moreover, crystalline HA has been proved to be biocompatible and osteoconductive, and it exhibited the slowest degradation rate compared with other calcium phosphates [29].

Modification of the functional properties (e.g., viscoelasticity, modulus, permeability, and injectability features) of nanocomposite hydrogels/gels has been employed for bone tissue regeneration [30,31,32]. Previous studies were also dedicated to developing 3D nanocomposite poly hydroxyapatite scaffolds for bone tissue regeneration [33]. In line with other studies, injectable hydroxyapatite hydrogel composites offered a better performance in guided bone regeneration [34,35]. A fibrous scaffold incorporated within a PEG hydrogel was shown to improve the mechanical properties [36]. In addition, a study revealed that collagen/n-HA hydrogels promoted spinal fusion [37]. A previous study evaluated the osteogenesis of bone marrow-derived human mesenchymal stem cells (hMSCs) with an implanted PEG-HA scaffold by an in vivo assay, showing that collagen deposition in the PEG-HA scaffold was prominent [8]. Additionally, a synthetic hydroxyapatite/β-tricalcium phosphate (HA/TCP) material blended with a PEG hydrogel, including a blend with human Bone Morphogenetic Proteins-2 (rhBMP-2), enhanced new bone tissue formation as compared to BMP-2 delivered using the HA/TCP construct alone [38].

Gold nanoparticles (AuNPs) offer a promising multi-angled approach for tissue regeneration [39]. AuNPs can easily be contoured to different sizes and shapes which can then be manufactured by surface plasmon resonance (SPR), a phenomenon involving the interaction between the electromagnetic wave and the conduction electrons in materials [40] which includes cytotoxicity, uptake, biodistribution, and pharmacokinetics [41]. As discussed in our previous reports, AuNPs have a distinct property which can affect many types of polymers such as polyurethane, fibronectin, and collagen via their interaction with the polymer to form the nanotopography, as well as the ability to display superior biological performance, including modifying cell behavior which is specific for induction involving the differentiation capability of stem cells [42,43,44]. AuNPs are appropriate components for the functionalization of electrospun scaffolds for bone regeneration, due to their properties of encouraging osteogenic differentiation in stem cells and osteoclasts [45,46,47,48]. The binding of AuNPs is specific for the surface of HA with its high binding affinity to HA [49]. Another report also confirmed that AuNPs can notably promote the osteogenic process in calvaria animal defect models [50]. Therefore, it has also been well established that AuNPs are potential nanoparticles for osteogenesis. Furthermore, whether AuNPs can enhance MSCs regarding osteogenic differentiation by upregulating the osteogenic gene Runt-related transcription factor 2 (Runx-2) [47] has been investigated.

Bone homeostasis is maintained by osteoblasts (OBs) and osteoclasts (OCs) within the basic multicellular unit, in a consecutive cycle of resorption and formation. Therefore, a functional scaffold should allow the best possible OB/OC cooperation for bone remodeling, as happens within the bone extracellular matrix in the body [51]. As a current treatment, bone grafts are associated with inherent limitations; hence, bone tissue engineering as an alternative therapeutic approach has been considered in recent decades. Through the concerted participation and combination of biocompatible materials, osteoprogenitor/stem cells and bioactive factors closely mimic the bone microenvironment. Bioactive factors regulate the cell behavior, and they induce osteogenic differentiation of stem cells by activating specific signaling cascades [52].

Mesenchymal stem cells (MSCs) secrete different types of growth factors and cytokines, which, in turn, promotes tissue regeneration and attracts stem cell-like progenitor cells for recruitment and differentiation [53]. Therefore, MSCs are believed to be an important stem cell for tissue engineering, as well as being capable of differentiating into various specialized tissues, including bone, tendons, cartilage, muscle, ligaments, and fat [54,55]. The fabrication of macroporous PEG hydrogel scaffolds for use as a bioactive material for human bone marrow-derived MSCs in osteogenic differentiation has been successfully reported [56]. Another report also demonstrated that amphiphilic poly (d, l-lactic acid)-co-poly (ethylene glycol)-co-poly (d, l-lactic acid) (PELA) with HA, fabricated by electrospinning, can lead to osteogenic differentiation of MSCs [57]. Furthermore, 3D MSC growth on PEG-HA biomimetic structures can enhance the polymerization process in the modulation of osteogenesis [58].

The purpose of this study was to establish a biomimetic nanostructured PEG-HA-AuNP substrate for improving material properties by having more hydrophilicity, biocompatibility, and anti-inflammatory capability, and establishing the osteogenic differentiation of MSCs, which were assessed through in vitro and in vivo assays.

## 2. Materials and Methods

### 2.1. Material Preparation

#### 2.1.1. Preparation of Polyethylene Glycol (PEG) Stock Solution

PEG 500 μM (Sigma-Aldrich, Burlington, MA, USA) was diluted 25 times to produce a final concentration of 20 μM. The material used in this experiment was prepared by applying the solution to a culture dish or 6-well, 24-well, or 96-well plates. After standing for 30 min, the coating solution was sufficiently brought into contact with the surface of the culture plate. The excess solution was then removed, becoming the material of the composite film to be used in the upcoming experiments.

#### 2.1.2. Preparation of Polyethylene Glycol-Hydroxyapatite (HA)

Hydrochloric acid was diluted to 2 N; then, 5 mL 2 N HCl was added to 0.1g HA and shaken to complete the dissolution. Afterwards, HA at a concentration of 20 mg/mL was diluted with PEG to obtain a PEG-HA solution (2.5 mg/mL). The mixing ratio was calculated from the mass conservation formula: M1V1 = M2V2 (M: concentration of solution; V: volume of solution). After coating for 30 min, the solution was sufficiently in contact with the surface of the glass. The excess solution was then removed and neutralized with 1 mL of 0.05% NaOH solution for 10 min, prior to being washed with phosphate buffer solution (PBS).

#### 2.1.3. Preparation of Polyethylene Glycol-Hydroxyapatite-Gold Nanoparticles (PEG-HA-AuNPs)

Gold nanoparticles (AuNPs) were purchased from Gold NanoTech Inc (Taipei, Taiwan). Gold NanoTech Inc utilizes unique and patented technology to physically break down gold into nanoparticles, followed by epitaxially stacking these nanoparticles into stacked materials with a controlled diameter within the nanometric range. Gold nanoparticles produced by this manufacturing process possess distinctive physical properties due to a special ionic charge that maintains their structure and are different from commercially available nanogold produced by chemical reduction methods. HCl was diluted to 2 N; then, 5 mL of 2 N HCl was added to 0.1 g of HA and shaken to the complete dissolution, at which time the HA concentration was 20 mg/mL. HA was then diluted with PEG (20 μM) to obtain an HA concentration in the PEG-HA solution of 2.5 mg/mL. Then, an AuNP solution at a concentration of 1.25 ppm was added to the PEG-HA solution, which was the working concentration used in the upcoming experiments. The mixing ratio was calculated from the mass conservation formula: M1V1 = M2V2 (M: concentration of solution; V: volume of solution). The material used in this experiment was prepared by applying the solution to a culture dish or 6-well, 24-well, or 96-well plates. After standing for 30 min, the coating solution was sufficiently brought into contact with the surface of the culture plate. The excess solution was then removed, becoming the material of the composite film to later be used in the upcoming experiments.

#### 2.1.4. Preparation of Surface Coatings

Different materials (PEG, HA, PEG-HA, PEG-HA-AuNPs, and PEG-AuNPs) were coated on the culture dish by applying the materials at the optimal concentration to cover the culture dish, plate, or 15 mm round coverslip glass. The coating solution was allowed to adsorb onto the surface of the culture area for 20–30 min. After coating, the residual content of the different materials was removed without any washing in order to establish a thin surface coating layer prior to cell culture.

### 2.2. Material Characterization

#### 2.2.1. Fourier Transform Infrared Spectroscopy (FTIR) Analysis

Fourier transform infrared spectrometry is derived from the Fourier transform, which requires the conversion of raw data into actual spectra. This experiment used potassium bromide (KBr) at 0.06 g (Sigma-Aldrich, Burlington, MA, USA), mixed with 200 mm of each experimental group, namely, PEG, PEG-HA, PEG-HA-AuNPs, and PEG-AuNPs. The scanning range was 600–4000 cm^−1^. A table change in surface functional groups was observed to demonstrate the different functional groups for each of the test materials [59].

#### 2.2.2. Surface-Enhanced Raman Scattering

The Raman examination was processed on a Raman microscope (LABRAM HR UVVIS-NIR Version) system, following our previously published report [59]. Materials were placed on a silicon substrate, and detection of the holographic grating with CCD was calculated over a total time of 60 s.

#### 2.2.3. UV–Visible Spectroscopy

A spectrophotometer measures the range of waves from 190 to 1100 nm, where the peak at 520 nm is the absorption wavelength of AuNPs. First, the quartz colorimetric tube must be cleaned with deionized water and wiped dry with mirror paper, and then the sample is added to the quartz colorimetric tube with deionized water for background absorption. After completion, each sample solution is measured sequentially. Prior to measuring the different samples, the quartz tube must be cleaned with deionized water to eliminate any residuals from the previous sample which could affect the absorption wavelength of the next sample. Origin Pro 8 (Originlab Corporation, Northampton, MA, USA) software analysis measured and quantified the data.

#### 2.2.4. Atomic Force Microscopy (AFM)

The material surface morphology was observed through atomic force microscopy (MFP-3D, Asylum Research, Santa Barbara, CA), as reported in our previous work [59]. Initially, 100 μL samples were cast on a Si (100) wafer substrate and then dried with blown air. Then, a silicon cantilever with a spring constant under 2.0 N constant conditions was used to observe the topography map of the materials. The resolution of the images processed in AC mode was 512 × 512 pixels. AFM results were obtained in three different experiments with samples at different scan areas.

#### 2.2.5. X-ray Photoelectron Spectroscopy (XPS)

The elemental composition of the materials was assessed by X-ray photoelectron spectroscopy (XPS) (JEOL JPS 9010 MX, JEOL Ltd., Akishima, Tokyo, Japan). The radiation source was emitted from Mg Kα X-rays [59].

#### 2.2.6. Free Radical Scavenging Ability

2,2-diphenyl-1-picrylhydrazyl (DPPH, Sigma-Aldrich, Burlington, MA, USA) was used to evaluate the free radical scavenging ability of AuNPs and polymer composites [60]. A control of distilled water (1 mL) or 1 mL of deionized water containing PEG and a PEG composite was added to 3 mL of DPPH in methanol and then left to stand for 90 min. The absorbance of the reaction mixture was then measured at 539 nm with an ultraviolet–visible spectrophotometer (Helios Zeta, Thermo, Waltham, MA, USA). The free radical scavenging ability effect is determined by the following equation: scavenging ratio (%) = [1 − (absorbance of test sample/absorbance of control)] × 100%.

### 2.3. Biocompatibility Test

#### 2.3.1. Cell Culture Condition

MC3T3-E1 Subclone 14 cells were obtained from ATCC. Cells were cultured in α-MEM culture medium (10% FBS, 100 U/mL P/S, 1% sodium pyruvate) and then removed with 0.05% trypsin-EDTA after reaching confluence at 37 °C, 5% CO_2_. MSCs were isolated from Wharton’s jelly tissue from the human umbilical cord [61]. Cells were cultured in a high-glucose DMEM culture medium (10% FBS, 100 U/mL P/S, 1% sodium pyruvate) and then removed with 0.05% trypsin-EDTA after reaching confluence at 37 °C, 5% CO_2_. For osteogenic differentiation, dexamethasone (0.1 μM, Sigma, USA) and ascorbic acid-2-phosphate (0.25 mM, Sigma, USA) were used.

For the characterization of MSCs, the specific surface markers of MSCs were characterized by flow cytometry. In brief, MSCs were detached, washed, and incubated with the indicated antibody conjugated with fluorescein isothiocyanate (FITC) and/or phycoerythrin (PE), against the indicated markers: CD14-FITC, CD34-FITC, CD44-PE, CD45-FITC, CD73-PE, and CD90-PE (BD Pharmingen, San Diego, CA, USA). PE-conjugated IgG1 and FITC-conjugated IgG1 were used as isotype controls (BD Pharmingen). Next, the antibody conjugated cells were analyzed by FACS analysis (LSR II, Becton Dickinson, Canton, MA, USA). MSCs in the 8th passage were used in this study. 

#### 2.3.2. MTT Assay

3-(4, 5-dimethylthiazol-2-yl)-2, 5-diphenyl tetrazolium bromide (MTT) was used as a substrate to react with mitochondrial dehydrogenase in living cells. Cells (1 × 10^4^/well) were cultured on a 96-well plate coated with PEG, PEG-HA, PEG-HA-AuNPs, and PEG-AuNPs after 24, 48, and 72 h of incubation. After incubation, cells were washed, and 100 μL of MTT reagent (0.5 mg/mL) was added to each well at 37 °C for 2–4 h. Afterwards, 100 μL of dimethylsulfoxide (DMSO) was added, and incubation was carried out for 10 min. The absorbance was read by an ELISA reader at 570 nm (Molecular Devices, SpectraMax M2, Molecular Devices, San Jose, CA, USA) [44].

#### 2.3.3. Reactive Oxygen Species (ROS) Generation Analysis

The substance test of the active peroxide was carried out using a fluorescent probe, DCFH-dA (2′,7′-dichlorofluorescin diacetate) (Sigma, USA). Cells (2 × 10^5^/well) were seeded in a 6-well plate coated with different materials at 37 °C for 48 h. After incubation, cells were collected, centrifuged, and washed with PBS. They were then stained with DCFH-dA (10 nM) at 37 °C for 30 min in dark conditions. The content of active oxidizing substances in the cells was calculated by a flow cytometer (BD, USA). Finally, ROS production was quantified using Flow Jo 7.6 (Becton Dickinson, Canton, MA, USA) software analysis [44].

#### 2.3.4. Actin Fiber Fluorescent Staining

Cells (1 × 10^4^/well) were seeded in a 24-well plate coated with different materials for 8 h and 24 h of incubation. The cells were then fixed with 4% paraformaldehyde (PFA) for 10 min and permeated with 0.5% Triton X-100 in PBS, and then they reacted for 10 min at room temperature. Afterwards, phalloidin (~6 μM) (Sigma, USA) was treated at room temperature for 60 min in dark conditions. Finally, 4, 6-diamidion-2-phenylindole (DAPI, (Invitrogen, White Plains, NY, USA) dye was diluted at a concentration of 1 μg/mL and allowed to stain the nucleus in the dark for 10 min at room temperature. Completing the process, coverslips were sealed with Gel Mount^TM^ (Sigma, USA), and the cytoskeletal patterns were observed under a fluorescent microscope [44].

#### 2.3.5. Cell Morphology and Adhesion Ability

Scanning electron microscopy (JEOL JEM-5200, JEOL Ltd., Akishima, Tokyo, Japan) was applied to observe the morphology and attachment ability of cells grown on different materials. Briefly, cells (1 × 10^4^/mL) were cultured for 48 h, fixed in a 2.5% glutaraldehyde solution for 8 h, and then dehydrated with ethanol at a concentration of 30% to 100%. Finally, after being dried at a critical point, the morphology of the cells on different materials was examined by SEM [44].

#### 2.3.6. Monocyte Activation Test

Whole blood was obtained from an adult human. The blood sample was diluted with PBS (1:1 ratio), Ficoll was added to it, and then it was centrifuged at 2000 rpm for 20 min. Afterwards, the plasma layer was removed and the buffy coat was collected. Cells (1 × 10^5^/well) were seeded in a 24-well plate coated with different materials for 96 h of incubation in an RPMI condition medium (10% FBS and 1% (*v*/*v*) antibiotics (10,000 U mL/penicillin G and 10 mg mL/streptomycin)). After incubation, cells were separated in the 24-well plate using 0.05% trypsin. Finally, the ratio of monocytes and macrophages was observed under a microscope. To further confirm the inflammatory response, CD68 (as a marker of macrophages) was also confirmed by immunofluorescence staining as described in a previous report [44].

#### 2.3.7. Platelet Activation Test

Platelets (2 × 10^6^ platelets/well) were cultured on different materials for a 24 h incubation period, fixed with a 2.5% glutaraldehyde solution for 8 h, and dehydrated with ethanol at a concentration of 30% to 100%. After 8 h, they were washed twice with PBS and dehydrated using a 30% to 100% alcohol concentration before the alcohol was removed after standing at room temperature for 10 min during each step. Finally, after being dried at a critical point, the morphology of the cells on different materials was examined by SEM (JEOL JEM-5200, USA), as described above [44].

### 2.4. Biological Functional Assay

#### 2.4.1. Cell Migration Assay

The following cell migration assay procedure was implemented as reported in our previous published report [44]. Briefly, cells (1 × 10^4^/well) were seeded in different materials through the use of a migration assay kit (Platypus Technologies, Madison, WI, USA) for a 96-well plate using a stopper. The stopper was removed, and incubation occurred for both 24 h and 48 h. Afterwards, Calcein AM (2 μM) was applied and stained for 30 min at each time point, prior to the cell migration ability being observed under a fluorescence microscope. The extent of cell migration was analyzed by Image J (National Institutes of Health, Bethesda, MD, USA) software.

#### 2.4.2. Gelatin Zymography Analysis

Cells (2 × 10^5^/well) were seeded into a 6-well plate overnight to allow the cells to attach. After 48 h of incubation, the culture medium was collected and determined as previously outlined. Samples were separated by 10% SDS-PAGE containing 2% gelatin, and the gel was incubated with a denaturing buffer (40 mM Tris-HCl, pH 8.5, 0.2M NaCl, 10 mM CaCl_2_, and 2.5% Triton X-100) for 30 min at room temperature. The gel was then slowly stirred at room temperature and equilibrated with a development buffer (40 mM Tris-HCl, pH 8.5, 0.2 M NaCl, 10 mM CaCl_2_, and 0.01% NaN_3_) for at least 16–18 h until it became activated in a 37 °C water bath. Afterwards, the gel was stained with 0.2% Coomassie Brilliant Blue R-250 (10% acetic acid and 50% methanol) and washed in a destaining buffer (10% acetic acid, 20% methanol). After Coomassie Blue staining, the protease-digested gelatin area appears as clear bands against a dark blue background. The resulting gel was digitized by scanning in a densitometer, and MMP gelatinase activity was quantified by Image Pro Plus 5.0 software (Media Cybernetics, Burlington, MA, USA).

#### 2.4.3. Enzyme-Linked Immunosorbent Assay

Murine macrophage RAW 264.7 cells were kindly provided by Professor Hui-Jen Chen and cultured in high-glucose Dulbecco’s modified Eagle’s medium supplemented with 10% fetal bovine serum, 100 μg/mL streptomycin, and 100 U/mL penicillin at 37 °C in 5% CO_2_. The cells were passaged every 2 days, and cells were seeded at the density of 1 × 10^5^ cells in each well (6-well plate). The cells were cultured in the different materials. After incubation for 12, 24, and 48 h, the medium was collected and then centrifuged at 210 g to obtain supernatants for an ELISA assay. The concentrations of TNF-α, IL-1β, IL-6, IL-10, and VEGF were measured with ELISA kits (R&D System) according to the manufacturer’s instructions. The concentrations of these cytokines were analyzed based on standard curves, and the results are shown as the amount (pg) of TNF- α IL-1β, IL-6, IL-10, and VEGF per ml of supernatant.

#### 2.4.4. Real-Time PCR Assay

The total amount of RNA in the cells was extracted by Trizol (lnvitrogen, Waltham, MA, USA). The method provided in the manufacturer’s manual was followed. Cells (1 × 10^5^/well) were seeded into a 10 cm culture dish after 7, 14, and 21 days of incubation. They were then treated with 1 mL of Trizol for 5 min, RNA was extracted by adding 200 μL chloroform (Sigma, USA) for 15 s, and then they were kept for 3 min at room temperature prior to being centrifuged at 12,000 rpm for 15 min at 4 °C. The supernatant was removed and 500 μL of isopropanol was added at 4 °C for 10 min. Finally, the samples were centrifuged at 12,000 rpm for 15 min at 4 °C. The supernatant was removed and washed twice with 1 mL of alcohol (75%). After drying the RNA, 20 μL of a DEPC-treated H_2_O-soluble precipitate was added and quantified by reading the absorbance at 260 nm using an ELISA reader (Molecular Devices, SpectraMax M2, USA).

cDNA synthesis was performed using a RevertAidTM First Strand cDNA DNA Synthesis Kit (Fermentas, Canada) following the manufacturer’s procedures. First, 2 μL of oligo (dT) 18 primer and random hexamers (1:1) were added to the RNA sample and placed in a gradient polymerase reaction temperature controller at 65 °C for 5 min. Then, the addition of 4 μL of 5× reaction buffer, 1 μL LockTM RNase inhibitor (20 U/mL), 2 μL dNTP Mix (10 mM), and 1 μL of RevertAidTM M-MuLV Reverse Transcriptase (200 U/mL) proceeded, before being reacted at 42 °C for 60 min. Finally, each sample underwent a reaction at 70 °C for 5 min to obtain cDNA.

The polymerase chain reaction was carried out using the cDNA as a template and a 1Q2 Fast qPCR System with a reaction volume of 10 μL according to the manufacturer’s procedures. Firstly, 0.5 μL of primer (0.3 μM) and 5 μL of enzyme were added to the cDNA sample, with the RNA expression then analyzed using the Step OneTM Plus Real-Time PCR System.

#### 2.4.5. Alizarin Red S Staining (ARS)

Cells (1 × 10^5^/well) were seeded into a 10 cm culture dish after 7, 14, and 21 days of incubation. After incubation, the cells were fixed with 4% PFA for 15 min and washed with PBS. A 2% Alizarin Red S staining solution was prepared and filtered, and the pH was adjusted to 4.1~4.3 so an Alizarin Red S working solution was made available. The cells were immersed in 500 μL of ddH_2_O for 1 min and then drained and removed. A 500 μL Alizarin Red S working solution was added after being reacted for 15 min at room temperature. Finally, it was drained and then soaked in 500 μL of deionized water for 2 min. The staining results were observed under a microscope and recorded.

### 2.5. Rat Subcutaneous Implantation

Sprague Dawley rats (300~350 g) were incised dorsally in an area of 10 mm^2^ under anesthesia prior to being subcutaneously implanted with the different materials. After 1 month of implantation, the samples were removed and examined using hematoxylin and eosin (H&E). We calculated the thickness of the fibrous capsule over 6 sites using Image J software to quantify the average encapsulated fibrotic tissues. The monoclonal anti-CD86, anti-CD163, and anti-CD45 antibodies (1:200 dilution) (Santa Cruz, CA, USA) were used to evaluate the activation levels of M1 and M2 macrophages. The secondary antibodies AF488 donkey anti-mouse IgG (1:500 dilutions) (Invitrogen, Carlsbad, CA, USA) and anti-mouse Immunoglobulin G (rhodamine) (1:500 dilution) (Jackson Immuno Research, West Grove, PA, USA) were used for signal detection. An Olympus ix71 fluorescence microscope (Tokyo, Japan) was then equipped to measure the fluorescence intensity. The amount of collagen deposition was observed as blue color using a Masson trichrome staining kit (Sigma, USA) according to the manufacturer’s instructions. The area of fibrosis tissue in the sample sections was calculated using Image J 4.5 version software (Media Gybertics). Three selected sections from randomly selected high-power fields (HPFs) were quantified for each animal and analyzed. After the number of pixels in each fibrosis area per HPF was measured, the number of pixels obtained from the three HPFs was collected and summed. The TUNEL assay is a procedure to detect DNA fragmentation by labeled 3′-hydroxyl termini in double-stranded DNA breaks which is generated during apoptosis. Paraffin-mounted heart sections from control and treated groups were deparaffinized with xylene and descending concentrations of ethanol. Apoptotic cells were stained based on the protocol provided by the In Situ Cell Death Detection Kit, AP (Roche Diagnostics, Indianapolis, IN, USA). DAPI solution was used to stain nuclei for investigation by a confocal microscope [62]. The number of rats was 5 (*n* = 5). Results are mean ± SD. All procedures followed the ethical guidelines and were approved by the animal care and use committee (La-1071565).

### 2.6. Statistical Analysis

Experiments were independently performed in triplicate to avoid uncertainty. Data for each test (*n* = 3–6) were collected and are presented as mean ± standard deviation. Student’s *t*-test and the single-factor analysis of variance (ANOVA) method were used to examine the differences between the groups. For ANOVA, Bonferroni was chosen for post hoc analysis. A *p* value less than 0.05 was considered to be statistically significant.

## 3. Results

The material preparation and surface morphology identification are illustrated in Figure 1A. The assembly mechanism of PEG-HA-AuNP nanocomposites may be attributed to the strong interaction between gold and oxygen atoms. This phenomenon implies that the additive interacts with the oxygen atoms in the PEG segment, causing the carbon atoms in the segment to be exposed to the surface. Indeed, it is also shown that the water contact angles of PEG, HA, PEG-HA, PEG-HA-AuNPs, and PEG-AuNPs were 24.9 ± 0.05°, 37.5 ± 0.45°, 17.8 ± 0.19°, 15.3 ± 0.46°, and 15.0 ± 0.22°, respectively (Figure 1B). The hydrophilicity property of materials is critical for the attachment of cells to the extracellular matrix (ECM) through cell adhesion molecules. In the present study, the addition of AuNPs to PEG-HA and PEG made the polymer surface more hydrophilic (Figure 1B), suggesting that the nanocomposites could facilitate the adhesion ability of MC3T3 cells and MSCs. These signals were clearly observed in PEG-HA and PEG-HA-AuNPs. UV–Vis spectroscopy was used to measure the absorption wavelength of the gold nanoparticles (AuNPs). The pure AuNPs have a typical peak at 520 nm and were observed in the PEG-HA-AuNP and PEG-AuNP groups (Figure 1C). As shown in Figure 1D, the specific peaks of PEG are 2931 cm^−1^ ν (-CH_2_)_asym_, 2868 cm^−1^ (CH_3_) _sym_, and 1105 cm^−1^ν (OH), with these three peaks being found in the PEG-HA, PEG-HA-AuNP, and PEG-AuNP groups, respectively, thus demonstrating the presence of HA and AuNPs in PEG. However, FTIR can only be used for the detection of organic functional groups and not for inorganic functional groups. Therefore, Raman spectroscopy was used to further confirm this result. As shown in Figure 1E, the specific peaks of polyethylene glycol are 1080 C-C, ν (CH_2_) cm^−1^, 1141 C-C, ν (CH_2_) cm^−1^, 1255 cm^−1^ ν (CH_2_) cm^−1^, 1297 ν (CH_2_) cm^−1^, 1482 cm^−1^ ν (CH_2_)_sym_ cm^−1^, 2892 cm^−1^, ν(CH_2_-CH_3_)_sym_ cm^−1^, and 2930 ν(CH_2_-CH_3_)_asym_ cm^−1^. The characteristic peaks of hydroxyapatite are 904 cm^−1^ (C_2_-H_deformation_) and 1664 cm^−1^ C=C amide I. Indeed, the free radical scavenging ability of PEG and PEG nanocomposites is demonstrated in Figure 1F. The capture capacity of PEG-HA was significantly higher than that of PEG or PEG-Au nanocomposites (*p* < 0.05). It is mainly the exposed HA on the surface that allows PEG-HA to combine with more free radicals. On the other hand, Au nanoparticles aggregated in the PEG polymeric chain, which led to a decrease in the free radical scavenging ability.

The surface morphology of pure PEG had a uniform and homogenous property, while the pure HA showed an irregular strip shape with rough protrusions. However, when PEG was mixed with the HA, it was observed that the surface morphology of the material became smoother and more homogenous. In addition, when PEG-HA was crosslinked with AuNPs, it prominently revealed that the surface shape of PEG-HA had generated bulk laminar-like protrusions with a homogeneous property. Specifically, when the AuNPs were added to the pure PEG matrix, the surface form was converted into a smaller strip-like nanostructure morphology (Figure 2). The surface roughness of PEG, HA, PEG-HA, PEG-HA-AuNPs, and PEG-AuNPs was 0.31 nm, 5.4 nm, 0.2 nm, 0.5 nm, and 0.97 nm, respectively (Figure 2A). This indicates that both PEG-HA-AuNPs and PEG-AuNPs had lower surface roughness conformity. Young’s modulus was used to analyze the material stiffness for this assay. The results for Young’s modulus show that PEG, HA, PEG-HA, PEG-HA-AuNPs, and PEG-AuNPs had Young moduli of 12 MPa, 550 MPa, 240 MPa, 130 MPa, and 6 MPa, respectively (Figure 2B). This shows that adding PEG and AuNPs into the HA matrix can cause the surface to become softer, particularly in the PEG-AuNP group. The stiffness of the material was investigated by measuring Young’s modulus in this study. The addition of gold nanoparticles to the PEG matrix did not have a significant effect on its stiffness upon statistical analysis. 

In addition, it was determined that when the incorporation of AuNPs into PEG-HA occurred, the surface morphology of PEG-HA-AuNPs was significantly changed according to SEM analysis (Figure 3A). This result is similar to the surface pattern of the AFM image (Figure 2A). Therefore, both PEG-HA-AuNPs and PEG-AuNPs have better hydrophilicity properties, which is beneficial for cell attachment. It was then confirmed that surface fabrication of PEG-HA by AuNPs can effectively induce pure PEG-HA substrates to generate different surface morphological changes. The XPS spectra of PEG and PEG composites are shown in Figure 3B. The atomic concentrations of O, C, Ca, P, and Au were calculated using XPS area measurement. The O1s/C1s ratio was 1.77 (Appendix A). When PEG contains gold nanoparticles or hydroxyapatite, the O1s/C1s ratio of the sample surface gradually decreases. However, only faint signals of gold nanoparticles were observed by XPS. This may be due to the strong interaction between gold and oxygen atoms. This phenomenon implies that the additive interacts with the oxygen atoms in the PEG segment, causing the carbon atoms in the segment to be exposed to thesurface.

The cell growth of MSCs and MC3T3 cells on PEG-HA-AuNPs after incubation for 48 h is shown in Figure 4A. The proliferation ability of MSCs at 48 h was the greatest on PEG-HA-AuNPs at 43.5 ppm, followed by 17.4 ppm and 174 ppm. Furthermore, ROS generation was the lowest in the concentration of PEG-HA-AuNPs at 43.5 ppm, followed by 17.4 ppm and 174 ppm, both in the MSCs and MC3T3 cells (Figure 4B). In addition, the CD68 fluorescence intensity on PEG-HA-AuNPs at 43.5 ppm was smaller than that on the other materials (Figure 4C). Semi-quantification data were established through CD68 immunofluorescence staining (*p* < 0.01) (Figure 4E). Moreover, platelets from the control group and control (glass) were almost flattened (activated form) (Figure 4D). In contrast, platelets from PEG platelets were less activated on all PEG-HA-AuNP nanocomposites, particularly at the concentration of 43.5 ppm, followed by 17.4 ppm and 174 ppm (*p* < 0.01) (Figure 4F).

We also characterized the MSC phenotypes by the detected surface markers of MSCs using FACS analysis (Appendix A). We detected negative surface makers such as CD14, CD34, and CD45, which were expressed in hematopoietic cells, endothelial cells, and immune cells, respectively, and positive surface antigens of MSCs: CD44, CD73, and CD90. Data from the FACS analysis further show that less than 2% of the negative markers (Appendix A) and higher than 98% of the positive markers were quantified (Appendix A) and validated in MSCs. 

The MTT results show that cell growth ability prominently increased in both PEG-HA-AuNPs and PEG-AuNPs when compared to the control group, particularly after 72 h in MC3T3 cells (*p* < 0.01). This stronger growth pattern was also observed in MSCs while being cultured on the different materials after 24, 48, and 72 h of incubation. This was seen specifically in the PEG-HA-AuNP and PEG-AuNP groups, followed by the PEG-AuNP, PEG-HA, and PEG groups, when compared to the control group (TCPS) (*p* < 0.01) (Figure 5A). Indeed, cells were cultured on different materials in order to investigate their intracellular ROS generation ability. While MC3T3 cell growth on different materials occurred over 48 h, the average amount of ROS production of PEG-HA-AuNPs (~0.69-fold) and PEG-AuNPs (~0.67-fold) in the cells was lower than that of PEG (~0.9-fold) and PEG-HA (~0.83-fold) when compared to the control group (*p* < 0.01) (Figure 5B). Indeed, a similar result occurred for MC3T3 cells while being cultured on different materials, as the anti-ROS generation ability was also observed after 24 h of incubation, specifically for the PEG-HA-AuNP and PEG-AuNP groups (*p* < 0.01) (Figure 5B). Similar to MC3TC cells, during MSC growth on different materials over 48 h, the ROS generation ability of both PEG-HA-AuNPs (~0.53-fold) and PEG-AuNPs (~0.47-fold) was lower than that of PEG (~0.9-fold) and PEG-HA (~0.89-fold) when compared to the control group (TCPS) (*p* < 0.01) after 48 h of incubation. Indeed, the average amount of ROS produced in the MSCs was lower in the PEG-HA-AuNP and PEG-AuNP groups than the other groups after 24 h of incubation (*p* < 0.01) (Figure 5B). Based on these results, it can be suggested that the PEG-HA-AuNP and PEG-AuNP groups had better antioxidant abilities, as well as good biocompatibility, in both MC3T3 cells and MSCs.

The actin fiber extension of MC3T3 cells and MSCs on different materials was observed through phalloidin staining. In one control group (glass), the cell shape was circular. However, the cytoskeleton was spread out to stress the fiber, particularly for the PEG-HA-AuNP and PEG-AuNP groups, followed by PEG-HA and PEG, in both MC3T3 cells and MSCs after 24 h of incubation (Figure 5C). At 24 h of incubation, the cell size of MC3T3 cells significantly increased in the control group (glass) (Figure 5C). Moreover, a similar pattern was also observed in MSCs. When MSCs attach or migrate on materials, they mainly produce lamellipodia and filopodia to change the morphology of cells (Figure 5C). Indeed, after 24 h of incubation, the cell sizes of MSCs in the control group, PEG, PEG-HA, PEG-HA-AuNPs, and PEG-AuNPs were 97.8 μm, 98.0 μm, 108.9 μm, 166.5 μm, and 155.9 μm, respectively. There was a significant difference in MSCs while being cultured in the PEG-HA-AuNP and PEG-AuNP groups, when compared to the control group (*p* < 0.01) (Figure 5D). It can be further inferred that both MC3TC cells and MSCs can promote cell attachment when cultured in nanomaterials. Moreover, this was also observed when cells cultured in either the PEG-HA-AuNP or PEG-AuNP group showed that the actin fiber elongation on MSCs was significantly higher than that on MC3T3 cells. The cell lengths of MC3T3 cells in the control (glass), PEG, PEG-HA, PEG-HA-AuNP, and PEG-AuNP groups were 107.8 μm, 130.7 μm, 129.3 μm, 182.6 μm, and 244.3 μm, respectively, after 8 h of incubation. Additionally, in MSCs, after 8 h of incubation, the cell lengths in the control, PEG, PEG-HA, PEG-HA-AuNP, and PEG-AuNP groups were 72.5 μm, 101.2 μm, 109.1 μm, 129.6 μm, and 146.6 μm, respectively. As shown in these results, both the PEG-HA-AuNP and PEG-AuNP groups significantly induced cell cytoskeleton change due to their production of more lamellipodia and filopodia, which can prominently induce cell actin fibers while exhibiting better extension properties. This, in turn, promotes a better adhesion and migration effect. Indeed, it was observed that the actin fiber extension of MC3T3 cells and MSCs while cultured on PEG-HA-AuNPs and PEG-AuNPs also had a similar extension tendency after 8 h of incubation when compared with the other groups (Appendix A). It can be further inferred that both MC3TC cells and MSCs can promote cell attachment when cultured in nanomaterials. Moreover, this was also observed when cells cultured in either the PEG-HA-AuNP or PEG-AuNP group revealed that the actin fiber elongation on MSCs was significantly higher than that on MC3T3 cells (Appendix A).

When an inflammatory response occurs, monocytes rapidly accumulate and differentiate into macrophages (~5 μm) within approximately 96 h (~40 to 45 μm) to trigger the immune response. The monocyte-to-macrophage transformation ratio on different materials after 96 h of incubation is presented in Figure 6A. The PEG-HA-AuNP and PEH-AuNP groups exhibited a lower monocyte activation effect and lower conversion yield compared to the control group (TCPS), followed by the PEG and PEG-HA groups. Thus, it was demonstrated that PEG-HA-AuNPs and PEG-AuNPs can promote anti-inflammatory abilities, as well as offering better biocompatibility towards avoiding any foreign body reaction. Indeed, the CD68 (an indicator of macrophage markers) fluorescence intensity on PEG-HA-AuNPs (~0.4-fold) and PEG-AuNPs (~0.33-fold) showed a lower expression than that of the PEG (~0.51-fold), PEG-HA (~0.51-fold), and control (TCPS) groups (*p* < 0.05). The semi-quantified data were calculated using Image J software (Figure 6C). The degree of platelet activation was observed through SEM analysis. As shown in Figure 6B, the number of adhered platelets was fewer, and they mostly represented a round morphology (non-activated form), while cell cultures in the PEG-HA-AuNP and PEG-AuNP groups did not. In contrast, regarding platelet cultures in the PEG, PEG-HA, and control groups, the adhered platelets increased and were subsequently exhibited as flattened (active form). The average degree of platelet adhesion ability in the PEG-HA-AuNP (~0.28-fold) and PEG-AuNP (~0.19-fold) groups was compared to the PEG (~0.99-fold), PEG-HA (~0.54-fold), and control groups (glass) (*p* < 0.01) (Figure 6D). Furthermore, the quantitative data also show that both PEG-HA-AuNPs and PEG-AuNPs can effectively suppress the number of monocytes and macrophages while also reducing the ratio of mononuclear cell conversion into macrophages (Appendix A).

The real-time images depicting cell migration were processed at both 24 and 48 h and observed under microscopy after Calcein AM staining. It was observed that with regard to the cell migration distance for MC3T3 cells (Appendix A) and MSCs (Appendix A) cultured on different materials, the PEG-HA-AuNP and PEG-AuNP groups were prominently higher than the PEG-HA, PEG, and control groups (TCPS) after 24 and 48 h of incubation. A similar result for MSCs while being cultured on different materials was also observed. The PEG-HA-AuNP and PEG-AuNP groups were prominently higher than the PEG-HA, PEG, and control groups (TCPS). Indeed, the quantification of the migration distance on MSCs cultured on the PEG-HA-AuNP (21 μm) and PEG-AuNP (23 μm) groups after 24 h and 48 h of incubation was prominently higher than that of MC3T3 cells cultured on the PEG-HA-AuNP (13 μm) and PEG-AuNP (15 μm) groups (*p* < 0.01) (Appendix A). Indeed, the migration distance of MSCs cultured on the PEG-HA-AuNP and PEG-AuNP groups was 28 μm and 32 μm, respectively, after 48 h of incubation; the migration distance of MC3T3 cells cultured on the PEG-HA-AuNP and PEG-AuNP groups was 31 μm and 32 μm, respectively (*p* < 0.01) (Appendix A). Based on these findings, it can be suggested that PEG-HA-AuNPs and PEG-AuNPs displayed a more prominent migration ability when compared to that of other materials. In particular, there was a significantly higher migration effect during the initial 24 h for MSCs. To explore whether MMP activity was induced by PEG-HA-AuNPs, both MC3T3 cells and MSCs were cultured on different materials. In the case of MC3T3 cells, the MMP-2 expression for PEG-HA-AuNPs and PEG-AuNPs was ~1.2-fold and ~1.24-fold, respectively, followed by PEG and PEG-HA, at ~1.08-fold and ~1.12-fold (*p* < 0.01), respectively. Additionally, the MMP-9 expression on PEG-HA-AuNPs and PEG-AuNPs was ~1.23-fold and ~1.28-fold, respectively, followed by PEG and PEG-HA, at ~1.14-fold and ~1.19-fold (*p* < 0.01), respectively, after 48 h of incubation on MC3T3 cells (Figure 6E). Indeed, the MMP-2 expression of MSCs on PEG-HA-AuNPs and PEG-AuNPs was ~1.26-fold and ~1.4-fold, respectively, followed by PEG and PEG-HA, at ~1.07-fold and ~1.12-fold (*p* < 0.01), respectively. Moreover, the MMP-9 expression on PEG-HA-AuNPs and PEG-AuNPs was ~1.28-fold and ~1.32-fold, respectively, followed by PEG and PEG-HA, at ~1.11-fold and ~1.18-fold, respectively, after 48 h of incubation on MSCs (*p* < 0.01) (Figure 6F). These findings indicate that PEG-HA-AuNPs and PEG-AuNPs may lead to higher MMP expression levels in MSCs than those in MC3T3 cells.

We found that osteogenesis induction of MSCs occurred after 7, 14 and 21 days of incubation on different materials. The semi-quantification data show that the PEG-HA-AuNP and PEG-AuNP groups were observed with slight mineral deposits as compared with the PEG, PEG-HA, and control groups (TCPS) on day 7. In addition, the mineral deposits increased in the PEG-HA-AuNP (~1.77-fold) and PEG-AuNP (~1.61-fold) groups when compared to the PEG (~1.00-fold), PEG-HA (~1.32-fold), and control groups (TCPS) over a longer period of time on day 21 (*p* < 0.01) (Figure 7A,B). Based on this finding, it can be suggested that PEG-HA-AuNPs and PEG-AuNPs may have the potential to enhance the differentiation capacity of MSCs into bone tissue. We also conducted real-time PCR analysis to investigate the Runx-2 (an osteogenic marker) gene expression level induced by different materials. In MSCs cultured on PEG-HA-AuNPs (~3.46-fold) and PEG-AuNPs (~3.58-fold), expression levels were significantly induced when compared with those for PEG (~2.05-fold), PEG-HA (~2.15-fold), and control group (TCPS) (*p* < 0.01) after 21 days of incubation (Figure 7C). The pattern of Runx-2 mRNA expression was similar to that taken from the observation of the ARS staining assay. In particular, PEG-HA-AuNPs and PEG-AuNPs had the potential to promote MSCs to differentiate into bone tissue. Osteocalcin (OCN), osteopontin (OPN), and alkaline phosphatase (ALP) were utilized as the osteoblastic differentiation markers. In our study, after induction of osteoblastic differentiation over 7, 14 and 21 days, real-time-PCR was performed for cells on different materials. PEG-HA-AuNPs showed the greatest upregulation of ALP at 2.26-fold, followed by OCN at 1.68-fold and OPN at 1.66-fold (Figure 7C). These results reveal that osteogenic marker genes in MSCs were upregulated by PEG-HA-AuNPs.

Bone vasculature plays an important role in bone development, remodeling, and homeostasis. Angiogenesis has been induced by the endothelial nitric oxide synthase gene through vascular endothelial growth factor expression in a rat ischemia model. Overall, intramuscular injection of the eNOS plasmid induced therapeutic angiogenesis in a rat ischemic hindlimb model, offering a potential therapy for peripheral arterial disease. The stimulation of angiogenesis caused by NO may be due to the upregulation of local VEGF expression [63]. We analyzed the VEGF expression of MSCs through an ELISA assay. After MSC growth on different materials over 48 h of incubation, VEGF expression was significantly induced by both PEG-HA-AuNPs (~1.37-fold) and PEG-AuNPs (~1.32-fold), as compared to the control group (*p* < 0.01) (Figure 8A). These results demonstrate that the nanocomposites may promote eNOS production and VEGF expression as inferred by the endothelialization capacity for bone tissue regeneration. We also investigated the macrophage-mediated cytokine expression levels on different materials. When RAW 264.7 cells were cultured on PEG-HA-AuNPs and PEG-AuNPs, the concentrations of IL-1β, IL-6, and TNF-α decreased over varying times of incubation. However, in the groups of PEG-HA-AuNPs and PEG-AuNPs, the levels of the anti-inflammatory cytokine IL-10 significantly increased compared to those in the other groups (Figure 8B). The schematic diagram shows that PEG-HA-AuNPs with MSCs induced better angiogenic and osteogenic differentiation. After combining with PEG-HA-AuNPs, the expression of CD 86 was decreased. In contrast, CD 163 expression was increased. This result indicates that PEG-HA-AuNPs may inhibit the inflammatory response. Moreover, PEG-HA-AuNPs effectively promoted endothelialization, leading to a higher expression of CD 31. They also induced the expression of the Runx-2 gene, which enhanced the differentiation of MC3TC cells into osteocytes. The former was linked to angiogenesis, while the latter was related to osteogenesis. The above evidence supports the theory that PEG-HA-AuNPs could become an outstanding biomaterial for bone tissue regeneration (Figure 8C).

The foreign body reaction from the different materials was determined through subcutaneous implantation over the course of one month to assess both the biocompatibility and inflammatory response. The characteristics of the fibrous capsule on both PEG-HA-AuNPs (~0.47-fold) and PEG-AuNPs (~0.47-fold) exhibited less thickness, followed by PEG (~0.81-fold) and PEG-HA (~0.91-fold), when compared to the control group (*p* < 0.01) (Figure 9A,D). Moreover, in order to further confirm the endothelialization ability of MSCs on PEG-HA-AuNPs in vivo, materials were subjected to subcutaneous implantation for one month. IHC staining showed that CD31-positive cells abundantly occurred with PEG-HA-AuNPs (~2.09-fold) and PEG-AuNPs (~1.69-fold), followed by PEG (~1.31-fold) and PEG-HA (~0.99-fold), when compared to the control group (*p* < 0.01). It can be suggested that PEG-HA-AuNPs displayed a superior endothelialization capacity when compared to the other test groups (Figure 9B,E). Furthermore, in order to confirm the foreign body reaction caused by different materials, we further stained the infiltration of CD45 (a marker of leukocytes) in order to assess the foreign body reaction to the materials after one month of subcutaneous implantation into rats. Indeed, the CD45 fluorescence intensity on PEG-HA-AuNPs (~0.32-fold) and PEG-AuNPs (~0.29-fold) showed a lower expression than that on the PEG (~ 0.8-fold), PEG-HA (~1.12-fold), and control (TCPS) groups (*p* < 0.01). The semi-quantified data were calculated using Image J software (Figure 9C,F). This result suggests that these materials may be safe for biomaterial application purposes.

Tissue samples were analyzed for collagen deposition and collagen concentrations as a measure of tissue fibrosis. Masson’s trichrome staining revealed significant collagen deposition in response to the control-treated group (glass). The intensity of collagen deposition in the PEG-HA-AuNP (~0.72-fold) and PEG-AuNP (~0.70-fold) groups was prominently lower than that in the PEG group (~0.92-fold), followed by the PEG-HA group (~1.45-fold), when compared to the control group (*p* < 0.01) (*p* < 0.01) (Figure 10A,D). Regarding this finding, the subcutaneous implantation of PEG-HA-AuNPs and PEG-AuNPs into rats reduced the amount of collagen deposition as well as attenuating the tissue fibrosis effect, particularly in the PEG-AuNP and PEG-AuNP-HA test groups. CD45 plays complex roles in both T cell and B cell antigen receptor signal transduction [64]. Subsequently, we further assessed the macrophage maker expression (M1: CD86, M2: CD163) in order to evaluate the inflammatory index of materials subcutaneously implanted for a period of one month. Indeed, CD86 expression was lower for PEG-HA-AuNPs (~0.37-fold) and PEG-AuNPs (~0.45-fold), followed by PEG-HA (~0.82-fold) and PEG (~0.69-fold), when compared to the control group (*p* < 0.01) (Figure 10B,E). Indeed, CD163 showed a more prominent expression on PEG-HA-AuNPs (~1.93-fold) and PEG-AuNPs (~1.81-fold), followed by PEG-HA (~1.15-fold) and PEG (~1.47-fold), when compared to the control group (*p* < 0.01). Based on this finding, it can be suggested that both PEG-HA-AuNPs and PEG-AuNPs may significantly promote the anti-inflammatory response in vivo (Figure 10C,F). The TUNEL test is an appropriate test for the biosafety concern surrounding these materials. The number of apoptotic cells was not obviously observed in all groups. There was not a significant difference in all test groups (Appendix A).

## 4. Discussion

Gold nanoparticles (AuNPs) have inert properties due to their excellent performance. As discussed in our previously published papers, when there is incorporation of an appropriate concentration of AuNPs into different types of polymers, leading to the formation of nanocomposites, it can cause superior biocompatibility and biological performance by nanotopography cues, as well as producing a biomimic interface, subsequently changing the cell behavior. The impressive capacity of nanocomposites may be attributed to their superior biocompatibility, which is not only due to their lower monocyte counts and platelet activation but also their higher cell growth, migration, and differentiation capabilities [59,65,66].

Hydrogels as stem cell scaffolds are a promising method for conquering initial cell loss and manipulating cell function post-transplantation. Scaffold degradation is required for downstream cell differentiation and functional tissue integration for determining results of therapies [67]. Additionally, electrospinning supplies various techniques for preparing micro- or nanoscopic fibers with matrices. Polymer materials which were produced through electrospinning had a high surface-to-volume ratio and pore interconnectivity [68]. The above research can be considered when seeking to provide a more effective and practical treatment and solution to bone tissue engineering. Polyethylene glycol (PEG) is modified into a variety of biomaterials in order to improve biocompatibility due to its superior safety and efficacy abilities [69]. MSCs are very useful in regenerative medicine due to their potential for the regeneration of different types of damaged tissues, particularly bone tissue [19,70]. Another study in the available literature illustrated that the differentiation of MC3T3-E1 pre-osteoblasts promotes growth in hydrogels through hydroxyapatite nanoparticles [71]. Higher adherence efficiency, as well as the induction of higher ALP activity, in human osteoblast-like MG-63 cells subsequently occurred. An HA-PEG composite involved chains of disuccinimidyl tartrate in order to obtain a PEG hydrogel. Therefore, HA-PEG composites offer potential as a bone substitute application [72].

Based on these findings, the use of PEG-HA-AuNP nanocomposites provides a microenvironment which mimics the microenvironment for differentiation into the bone tissue of MSCs. The elastic properties of the matrix correlate with stem cell differentiation (Figure 2), where they can guide different cell behavior changes via mechanical attraction cues. Moreover, the surface roughness of the original PEG and PEG-HA was approximately ~0.31 nm and ~5.4 nm, respectively, with the roughness of PEG-HA-AuNPs and PEG-AuNPs significantly reduced to ~0.5 and ~0.97 nm, respectively, after the addition of an optimal concentration of AuNPs (Figure 2). Therefore, it can be inferred that an appropriate concentration of AuNPs can significantly improve the roughness of the PEG-HA material surface to become smoother and more uniform. After performing calculations, the Young modulus values for PEG-HA-AuNPs and PEG-AuNPs were lower than those for PEG and PEG-HA (Figure 2), demonstrating that the properties of PEG-HA-AuNPs and PEG-AuNPs were softer and had decreased through the incorporation of an optimal concentration of AuNPs, as seen in the AFM analysis. Based on the above results, it can be suggested that the specific nanotopography features of both PEG-HA-AuNPs and PEG-AuNPs can be effective in guiding cells towards producing better attachment, growth, migration, and differentiation. Meanwhile, much published literature has proved that most cells prefer to adhere to the hydrophilic surface of biomaterials [18]. Our results found that when there was cell growth on PEG-HA-AuNPs and PEG-AuNPs, the hydrophilicity properties significantly increased when compared to both the PEG-HA and control (TCPS) groups (Figure 5). Until now, much research has indicated that when stem cells are in the differentiation stage, the activation of MMP2/9 will be triggered and therefore effectively promote stem cells towards migration and differentiation events through SDF-1α/CXCR4 signaling pathways [53]. Interestingly, we also observed the migration rate of MSCs and saw better migration efficiency when compared to MC3T3 cells after 24 h of incubation, particularly for the PEG-HA-AuNP and PEG-AuNP groups. Therefore, we deduced that nanocomposites can stimulate MMP expression, which, in turn, may promote superior differentiation for the osteogenesis of MSCs.

Gold-hydroxyapatite nanostructures potentiate the osteogenic differentiation of MSCs, as evidenced by a prominent over-expression of Runt-related transcription factor 2 (Runx-2) and Bone Morphogenetic Protein 2 (BMP-2) genes after 7 days of incubation [73]. As evidenced from this result, PEG-HA-AuNPs and PEG-AuNPs can significantly induce Runx-2 gene expression (Figure 7), as well as producing the osteogenesis of MSCs after being cultured on materials for 7, 14 and 21 days of incubation (Figure 7). Moreover, the incorporation of AuNPs, which seemed to stimulate the osteoinductive capability of HA-AuNP nanomaterials, may modulate the Wnt/β-catenin signaling pathway [74]. Therefore, elucidating the molecular mechanism of PEG-HA-AuNPs and PEG-AuNPs has the potential to promote superior biocompatibility, biological capability, and differentiation capacity, although this still requires further exploration in the future.

It is well known that angiogenesis plays an important physiological function in the process of bone tissue regeneration [71]. New blood vessel formation is essential during both primary bone development and fracture repair in adults. Both bone repair and remodeling involve the activation and complex interaction between angiogenic and osteogenic pathways. Interestingly, studies have demonstrated that angiogenesis precedes the onset of osteogenesis [75]. We have previously demonstrated that the addition of gold nanoparticles in polyurethane can induce changes in nanomorphology and effectively activate the expression of eNOS and VEGF, thereby promoting vascular endothelialization, angiogenesis, and endothelial tissue repair [65]. Bone vasculature plays a vital role in bone development, remodeling, and homeostasis. New blood vessel formation is crucial during both primary bone development and fracture repair in adults. Both bone repair and bone remodeling involve the activation and complex interaction between the angiogenic and osteogenic pathways. Angiogenesis precedes the onset of osteogenesis [75]. The stimulation of angiogenesis by NO may be due to the upregulation of local VEGF [61]. In order to better understand whether the materials had a promoting effect on the production of eNOS by endothelial cells, we plated endothelial cells on the materials for 48 h and found that PEG-HA-AuNPs and PEG-AuNPs significantly increased eNOS production and promoted endothelialization (Figure 8). From in vivo animal experiments, we discovered that after materials were subcutaneously implanted into rats for a period of one month, the PEG-HA-AuNP and PEG-AuNP groups effectively promoted a higher CD31 expression level (as an endothelial marker) than that in the PEG and PEG-HA groups (Figure 9). Additionally, successfully inhibited lipopolysaccharide (LPS)-induced interleukin-6 (IL-6) and inducible nitric oxide synthase (iNOS) secretion by M1 macrophages facilitates osteogenesis, which was investigated on a rat calvarial defect model [76]. Therefore, in order to prove whether PEG-HA-AuNP nanocomposites can be safely used in the human body, we further detected the foreign body reaction (FBR) through HE staining. As shown in this study, both the PEG-HA-AuNP and PEG-AuNP groups were able to effectively reduce the production of capsules after subcutaneously implanting materials for a period of one month (Figure 9).

Several studies have shown that pro-inflammatory cytokines are accompanied by the sensitization of cells such as neutrophils, macrophages, and endothelial cells. RAW 264.7 cells are commonly used as macrophages to show phenotype alterations in response to environmental changes [77,78]. Thus, we selected RAW 264.7 cells in the present study to investigate inflammatory cytokine secretion in different materials (Figure 8). Additionally, we also doubly confirmed the anti-inflammatory response of nanocomposites through IHC staining. The activation of CD86 (as a marker of M1) was decreased, while the expression of CD163 (as a marker of M2) was increased in both the PEG-HA-AuNP and PEG-AuNP groups. Based on these findings, we suggest that PEG-HA-AuNPs and PEG-AuNPs potentially inhibited the inflammation process and anti-fibrosis ability in vivo (Figure 10).

One study has pointed out that converting collagen and HA into bone biomaterials can effectively promote bone tissue mineralization, as well as inducing the regeneration of bone when there is an addition of iron and manganese nanoparticles [79]. Similar to this finding, our research also demonstrates that the modification of HA with PEG and the incorporation of an optimal concentration of AuNPs not only effectively promote the potential of stem cells to differentiate into bone tissue but also display a strong biocompatibility ability and superior functionality. Indeed, another published report also demonstrated that the modification of Ti biomaterials with gold nanoparticles can effectively promote bone regeneration in a rabbit model [80]. Meanwhile, a novel triblock PEG-PCL-PEG copolymer composed of collagen and nano-hydroxyapatite (n-HA) (PECE/Collagen/n-HA) offered better biocompatibility as well as superior performance in osteogenesis, leading to a bone tissue healing process in New Zealand white rabbits after being implanted over 4-, 12-, and 20-week periods [35]. In relation to this finding, our research also found that incorporating AuNPs into PEG-HA can effectively promote MSCs to differentiate into bone tissue. Therefore, we will use this PEG-HA-AuNP nanocomposite as a model system to further confirm whether PEG-HA-AuNPs also provide better repair potential for bone tissue regeneration in vivo in our future work.

Regarding these findings, it can be suggested that the PEG-HA-AuNP and PEG-AuNP groups were able to provide a suitable biocompatibility property for cell attachment, proliferation, and proper differentiation. The findings also prove that the addition of an optimal concentration of AuNPs onto the PEG-HA material surface significantly promoted the efficiency of osteogenic differentiation. Our results reveal that, due to their proper biocompatibility, as well as superior biological capability, PEG-HA-AuNPs and PEG-AuNPs may be considered as ideal nano-biomaterials and further applied towards the repair of bone tissue in the future.

## 5. Conclusions

In this report, we outlined how a concise surface modification technique was applied to produce PEG-HA-AuNP nanocomposites. This approach successfully constructed a PEG-HA-AuNP model, in which eminent materials were optimized to achieve advanced biocompatibility and biological capability, as observed in the in vitro assay. This study also illustrated that PEG-HA-AuNPs can induce MSCs via MMP activation and subsequently trigger a migration effect, as well as guiding the osteogenic differentiation of MSCS on this material. Simultaneously, the PEG-HA-AuNP nanocomposites effectively improved the endothelialization capability, anti-inflammatory response, and anti-fibrosis ability after subcutaneous implantation of PEG-HA-AuNPs into a rat after 1 month, making them a potential candidate for surface modification of biomaterials in bone tissue regeneration.

## Figures and Tables

**Figure 1 biomedicines-09-01632-f001:**
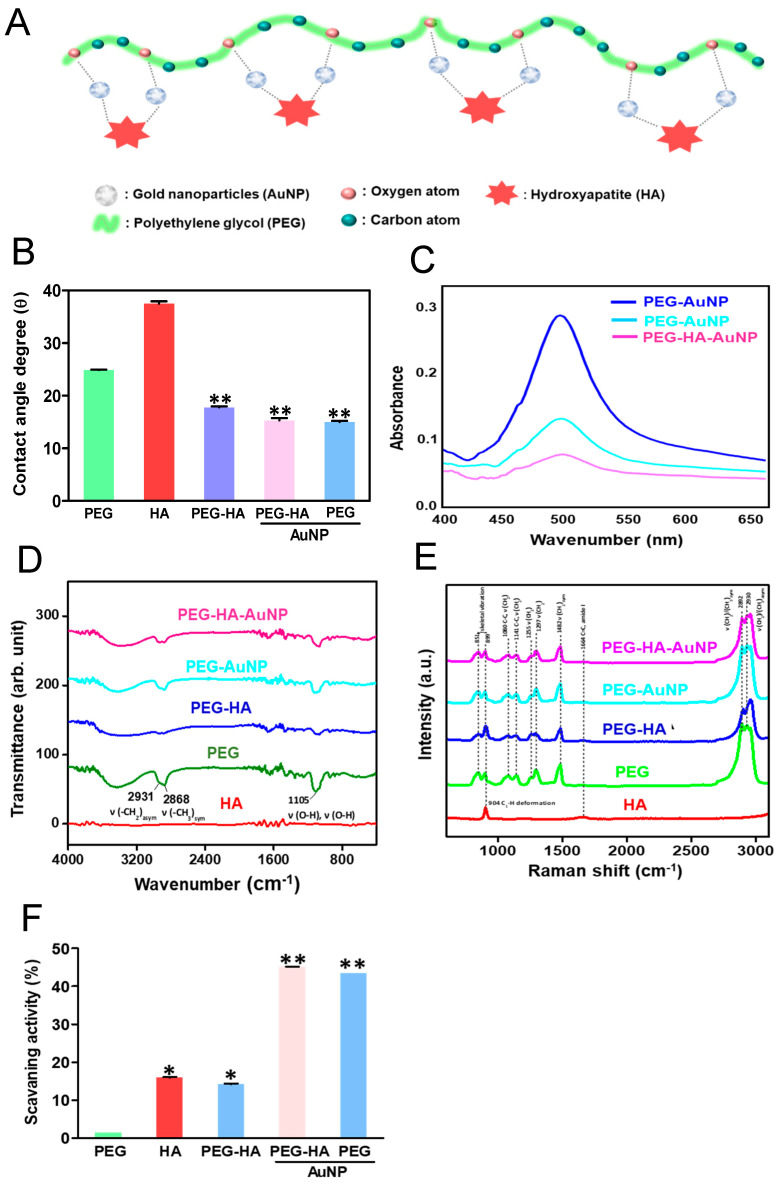
(**A**) Schematic diagram illustrating the preparation procedure of PEG-HA-AuNPs. Characterization of PEG-HA-AuNP nanocomposites: (**B**) Average contact angle (θ) quantified from different materials. The contact angle from different materials without water is θ = 0°. Data are the mean ± SD (*n* = 3), ** *p* < 0.01, smaller than the control treatment (PEG). (**C**) UV–Vis spectra, (**D**) FTIR spectra, and (**E**) Raman spectra of different materials. (**F**) Free radical scavenging effect of HA, PEG, and PEG nanocomposites. Free radical scavenging effect of HA, PEG, and PEG nanocomposites. Mean ± SD.* *p* < 0.05; ** *p* < 0.01: greater than control.

**Figure 2 biomedicines-09-01632-f002:**
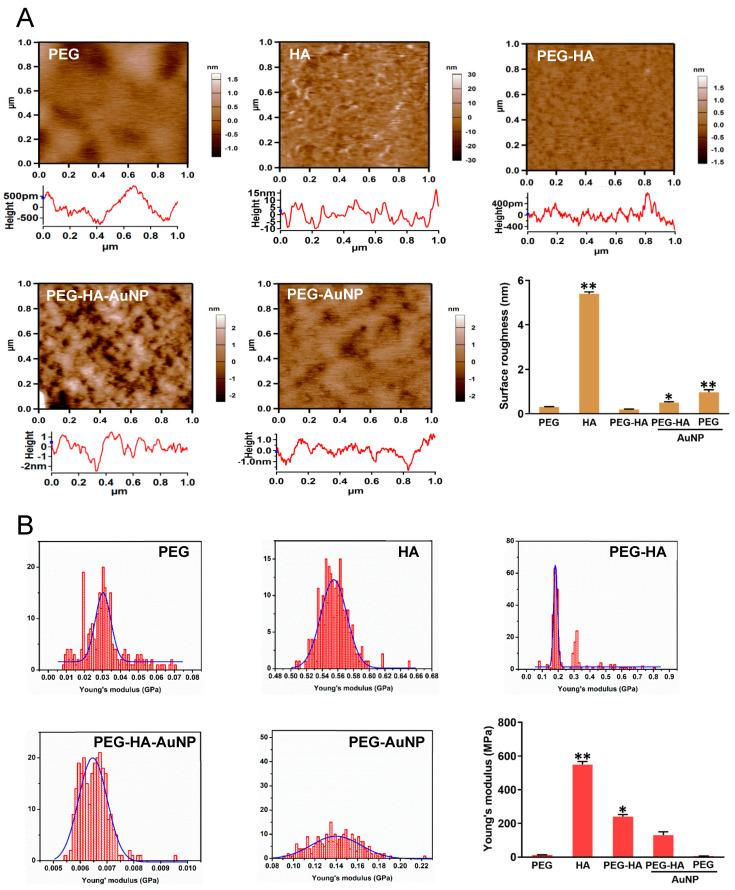
Surface roughness property characterization of different materials by AFM analysis. AFM images of different materials. (**A**) Topographical images of different materials. (**B**) Histograms of Young’s modulus values (MPa) of different materials. The quantification data of surface roughness of AFM, and the quantification of Young’s modulus of different materials. * *p* < 0.05; ** *p* < 0.01: smaller than control.

**Figure 3 biomedicines-09-01632-f003:**
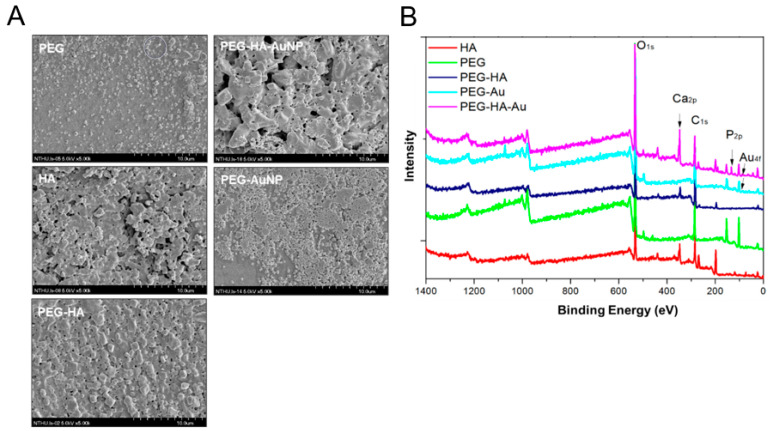
Material characterization. (**A**) SEM analysis of different materials. (**B**) Wide-scan spectra of PEG and PEG composites by XPS analysis.

**Figure 4 biomedicines-09-01632-f004:**
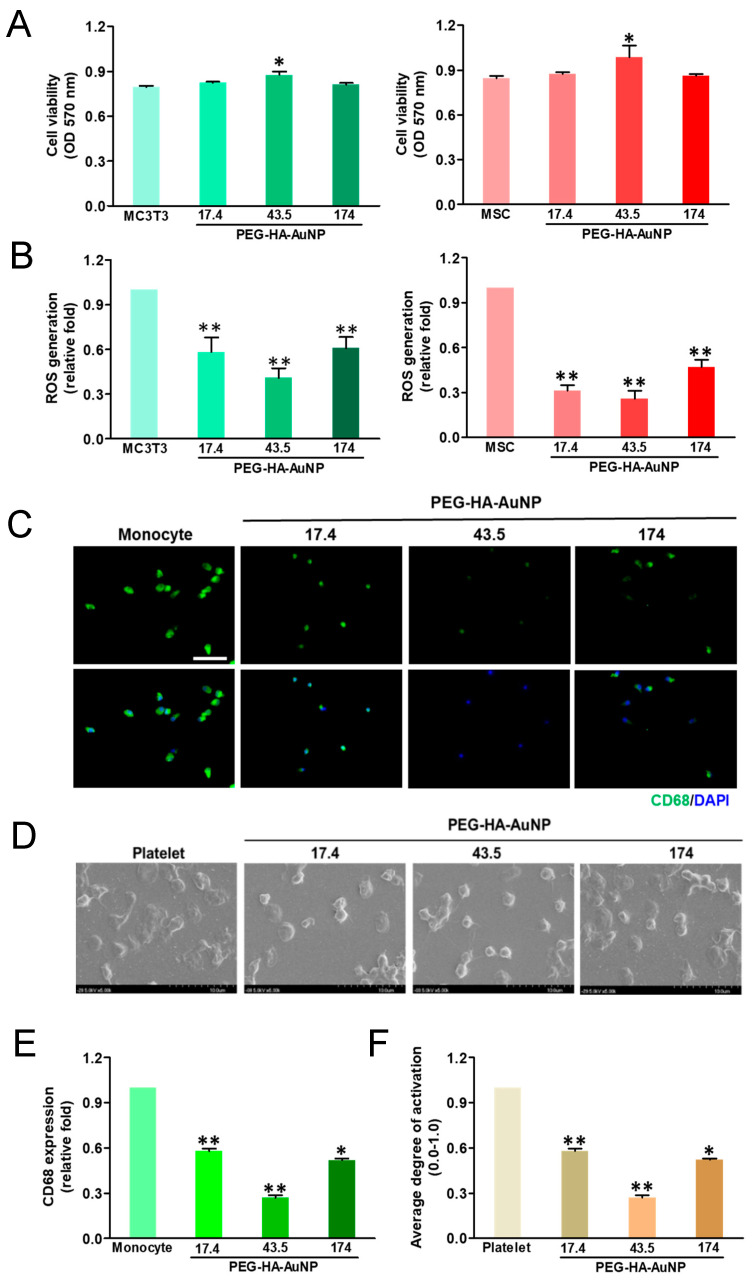
Cell proliferation of (**A**) MC3T3 cells and MSCs was promoted by culture on TCPS and different concentrations of AuNPs in the PEG-HA matrix. Data are the mean ± SD (*n* = 3), * *p* < 0.05; ** *p* < 0.05. Reactive oxygen species (ROS) generation assay of (**B**) MC3T3 cells and MSCs on different materials after 48 h of incubation. Intracellular ROS quantified by 2,7-dichlorofluorescein diacetate (DCFH-dA) and flow cytometric analysis. * *p* < 0.05; ** *p* < 0.01: smaller than control (TCPS). Biocompatibility assay. (**C**) The expression of CD68 for macrophages on different materials at 96 h. Cells were immunostained by the primary anti-CD68 antibody and conjugated with FITC-immunoglobulin secondary antibody (green color fluorescence). Cell nuclei were stained by DAPI (blue color fluorescence). Scale bar = 20 μm. (**D**) SEM images showing the adhesion and activation of human blood platelets on different materials. (**E**) CD68 expression was quantified based on fluorescence intensity. * *p* < 0.05: smaller than control (TCPS). (**F**) Quantification of the degree of platelet activation score. Data are the mean ± SD (*n* = 3). * *p* < 0.05; ** *p* < 0.01: smaller than control (TCPS). Based on these findings, we thus chose PEG-HA-AuNPs at 43.5 ppm in the following experiments.

**Figure 5 biomedicines-09-01632-f005:**
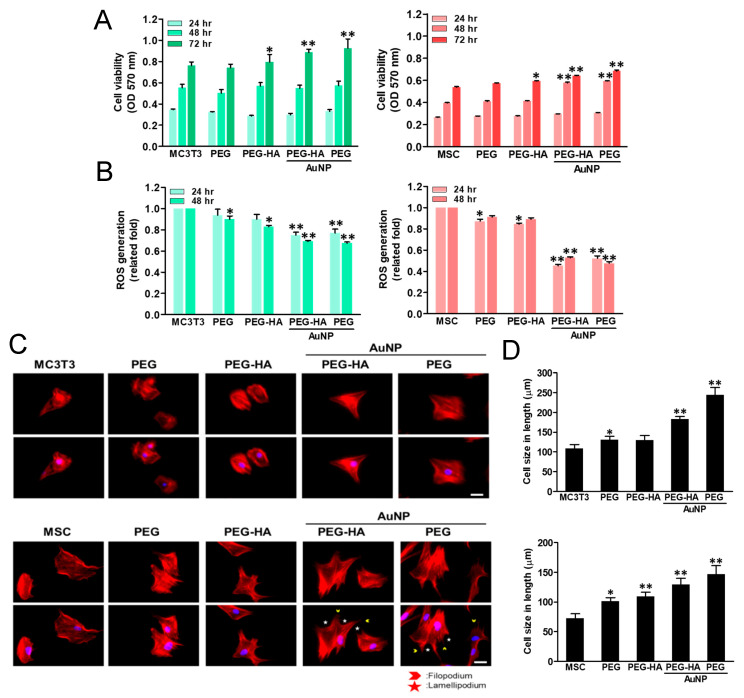
Cell proliferation of (**A**) MC3T3 cells and MSCs was promoted by culturing them on PEG-HA-AuNPs and PEG-AuNPs. Data are the mean ± SD (*n* = 3), * *p* < 0.05; ** *p* < 0.05. Reactive oxygen species (ROS) generation assay of (**B**) MC3T3 cells and MSCs on different materials after 48 h of incubation. Intracellular ROS quantified by 2,7-dichlorofluorescein diacetate (DCFH-dA) and flow cytometric analysis. * *p* < 0.05; ** *p* < 0.01: smaller than control (TCPS). Cytoskeleton and cell morphology by rhodamine phalloidin staining of (**C**) MC3T3 cells and MSCs for actin fiber extension on different materials after 24 h of incubation under fluorescence microscopy analysis. Scale bar = 20 μm. Arrows indicate filopodia (green color) and lamellipodia (red color). Data are the mean ± SD (*n* = 3). Actin fiber extension in length quantified by Image J software in (**D**) MC3T3 cells and MSCs on different materials after 8 h is shown. Actin fiber length elongation was significantly observed in the PEG-HA-AuNP and PEG-AuNP test groups compared with the other groups. Data are mean ± SD (*n* = 3). * *p* < 0.05; ** *p* < 0.05. Scale bar = 50 μm.

**Figure 6 biomedicines-09-01632-f006:**
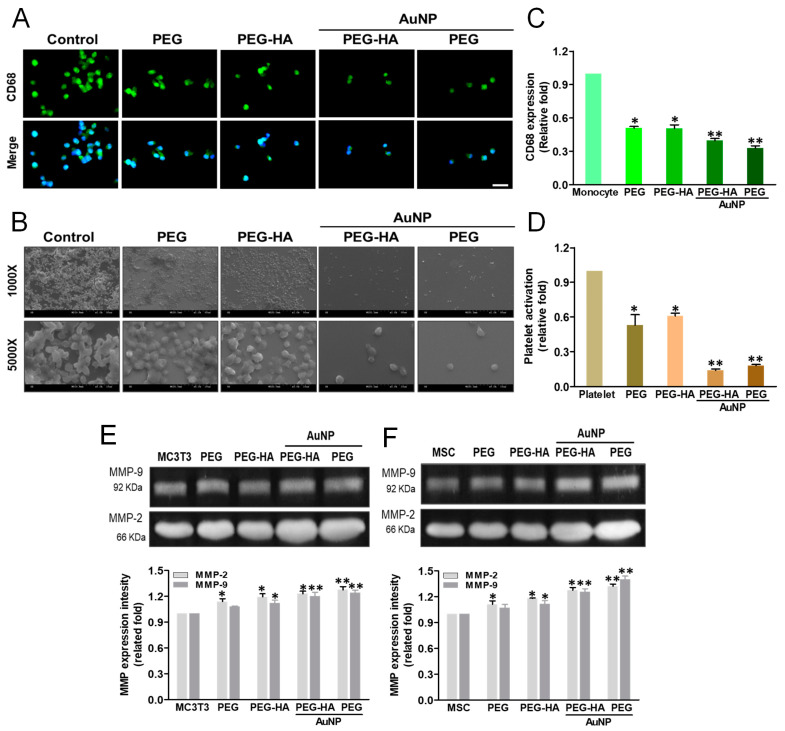
Biocompatibility assay. (**A**) The expression of CD68 for macrophages on different materials at 96 h. Cells were immunostained by the primary anti-CD68 antibody and conjugated with the FITC-immunoglobulin secondary antibody (green color fluorescence). Cell nuclei were stained by DAPI (blue color fluorescence). Scale bar = 20 μm. (**B**) SEM images showing the adhesion and activation of human blood platelets on different materials. (**C**) CD68 expression was quantified based on fluorescence intensity. * *p* < 0.05: smaller than control (TCPS). (**D**) Quantification of the degree of platelet activation score. Data are the mean ± SD (*n* = 3). * *p* < 0.05; ** *p* < 0.01: smaller than control (TCPS). The MMP-2/9 enzymatic activity of (**E**) MC3T3 cells and (**F**) MSCs was increased in the PEG-HA-AuNP and PEG-AuNP groups. Semi-quantitative measurement of MC3T3 cells and MSCs shows the expression level of MMP-2/9 protein for cells on different materials after 48 h of incubation. Semi-quantitative data in the graph represent the optical density (OD) of gelatinolytic bands. Data are the mean ± SD (*n* = 3). * *p* < 0.05; ** *p* < 0.01: greater than control (TCPS).

**Figure 7 biomedicines-09-01632-f007:**
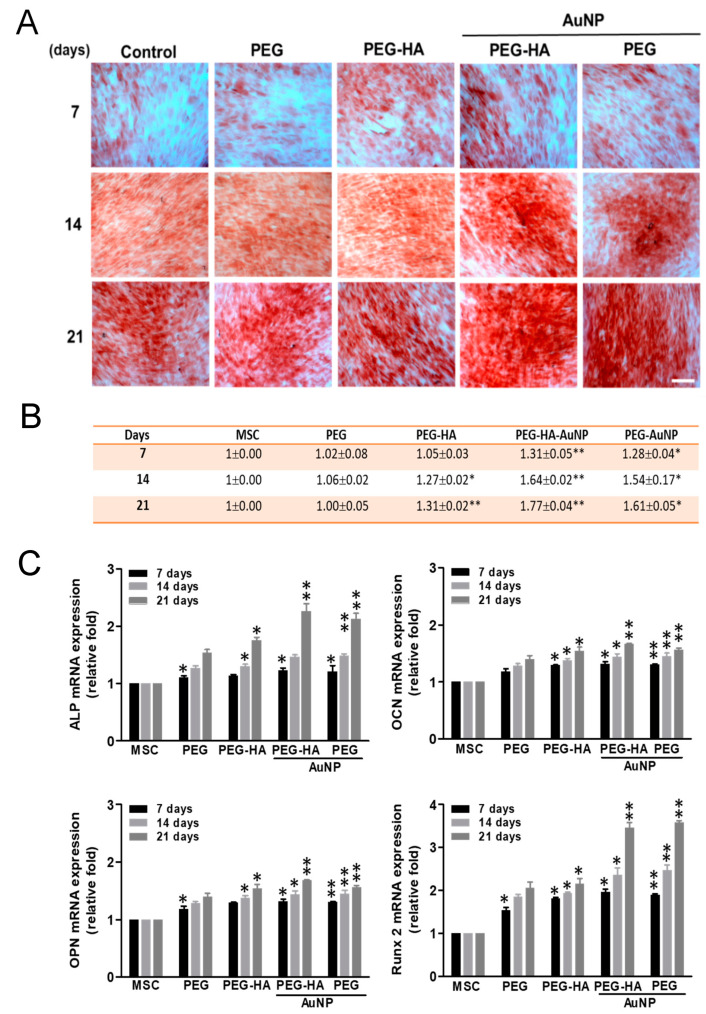
Osteogenic differentiation. (**A**) Osteogenic differentiation was confirmed by ARS staining of MSCs on different materials after 7, 14 and 21 days of incubation. Scale bar = 20 μm. (**B**) Semi-quantification of osteoblastic differentiation by ARS staining. All images represent the mean ± SD of three independent experiments. Data are the mean ± SD (*n* = 3). * *p* < 0.05; ** *p* < 0.01: greater than control (TCPS). (**C**) Real-time PCR analysis of mRNA expression level for Runx-2, ALP, OCN, and OPN in MSCs after being cultured with different materials. GAPDH was used as an internal control. The results are represented as the ratio of Runx-2/GAPDH signals for each condition, normalized to control. Data are the mean ± SD (*n* = 3). * *p* < 0.05; ** *p* < 0.01: greater than control (TCPS).

**Figure 8 biomedicines-09-01632-f008:**
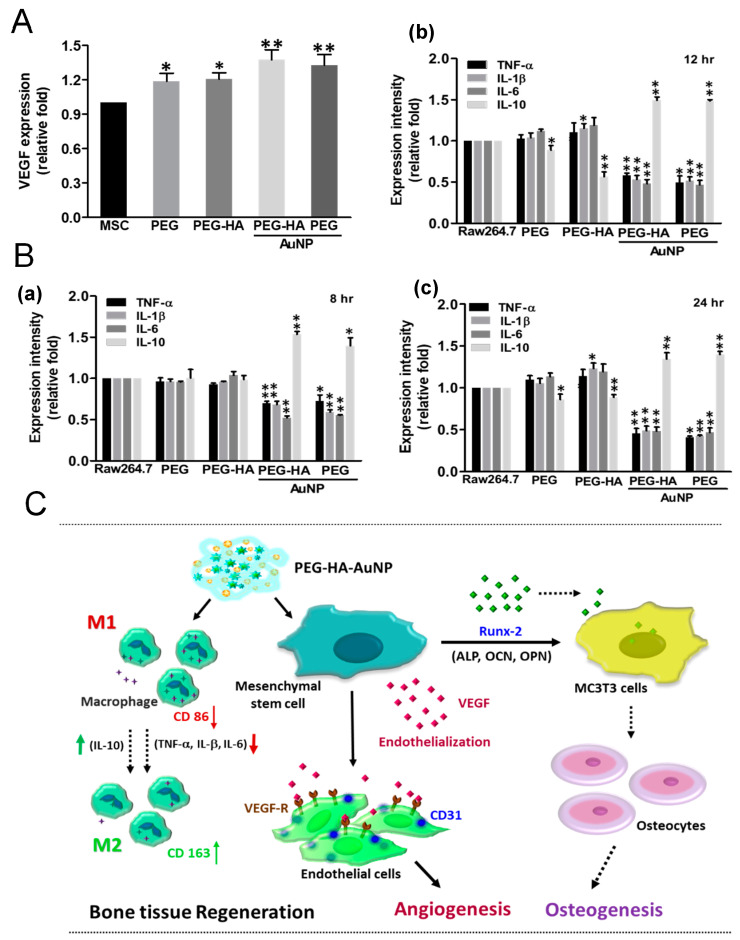
Assessment of the immune response and angiogenesis of MSCs on different materials. (**A**) Measurement by ELISA assay shows significantly increased VEGF protein expressions for MSCs after culture on different materials after 48 h of incubation. Data are mean ± SD. * *p* < 0.05: ** *p* < 0.01: greater than TCPS. (**B**) ELISA results for inflammatory cytokines: (**a**) 8 h, (**b**) 12 h, and (**c**) 24 h of production by RAW264.7 cells. * *p* < 0.05; ** *p* < 0.01. (**C**) Scheme illustrates PEG-HA-AuNPs’ prominent superior biological and biocompatibility performance, which may account for the induction of the better differentiation ability into bone tissue of MSCs for this substrate. Schematic diagram shows that PEG-HA-AuNPs with MSCs induced better angiogenic and osteogenic differentiation. After combining with PEG-HA-AuNPs, the expression of CD 86 was decreased. In contrast with CD 86, CD 163 expression was increased. This result indicates that PEG-HA-AuNPs could inhibit the inflammation response. Moreover, PEG-HA-AuNPs effectively promoted endothelialization, leading to a higher expression of CD 31; they also induced the expression of the Runx-2 gene, which caused MC3T3 cells to differentiate into osteocytes. The former was linked to angiogenesis, and the latter was related to osteogenesis. The above evidence shows that PEG-HA-AuNPs could become an outstanding biomaterial for bone tissue regeneration.

**Figure 9 biomedicines-09-01632-f009:**
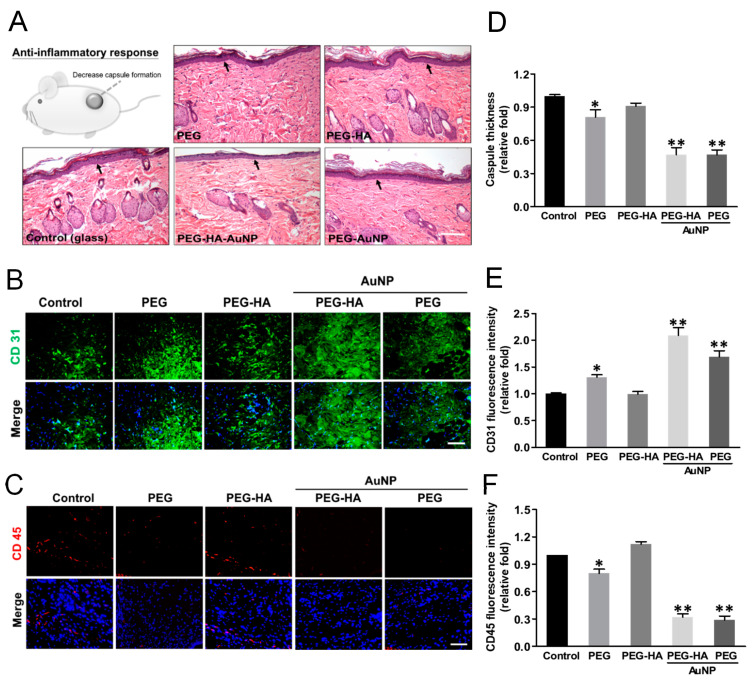
Evaluation of foreign body response of different substrates after subcutaneous implantation. Histology of H&E-stained sections and IHC staining after implantation of material for 4 weeks. (**A**) FBR is exhibited by the capsule thickness (arrows) based on the histology examination. The scale bar is 100 μm. (**B**) Immunofluorescence staining of CD31 (marker of endothelialization), and (**C**) CD45 (marker of immunoflammation) in response to the implant materials. The scale bar is 100 μm. Quantification of (**D**) capsule thickness, and (**E**) CD31 and (**F**) CD45 fluorescence intensities. Data are mean ± SD. * *p* < 0.05, ** *p* < 0.01: greater than control (glass). The number of rats was 5 (*n* = 5).

**Figure 10 biomedicines-09-01632-f010:**
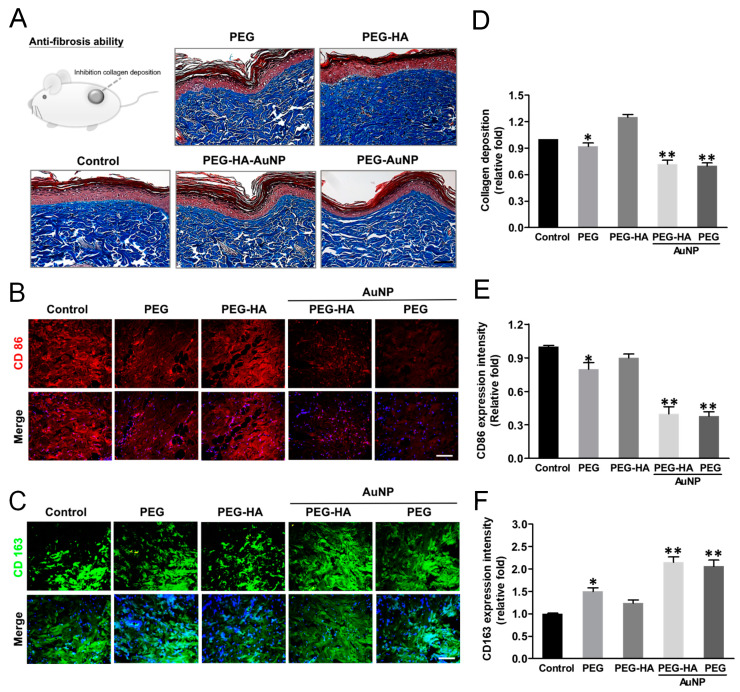
Masson’s trichrome staining of (**A**) collagen deposition (blue color) of rat femoral artery subcutaneously implanted in an SD rat assessed the immune response of different materials at 4 weeks. Immunofluorescence staining images (marker of macrophages) in response to the implant materials. (**B**) CD86 (M1) = red color, (**C**) CD163 (M2) = green color. The scale bar represents 100 μm. Quantification of (**D**) collagen deposition, and (**E**) CD86 and (**F**) CD163 fluorescence intensities. Data are mean ± SD. * *p* < 0.05, ** *p* < 0.01: smaller or greater than control group (glass). The number of rats was 5 (*n* = 5).

## Data Availability

Data are contained within the article.

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
