# Peer review of "Physical Gold Nanoparticle-Decorated Polyethylene Glycol-Hydroxyapatite Composites Guide Osteogenesis and Angiogenesis of Mesenchymal Stem Cells"

_biomedicines, 2021, doi:10.3390/biomedicines9111632_

Round 1
Reviewer 1 Report
This is an excellent research paper!! Very important for osteogenesis and angiogenesis research works.
However I have following suggestions:
Please use Italicized texts wherever required.
2.1.1. Preparation of PEG: PEG (500 mm) diluted 25 times and final concentration reaching 20 micro Molar. Please check and correct the unit.
Line 231, 234: •100 U/ ml. Please check the •.
2.3.4. Please mention the manufacturer of phalloidin and DAPI, mounting gel.
Line 272, 290. At a concentration of 30% to 100%. Please mention percentage of what.
Line 325, 100 ug/m streptomycin. Please make it ml.
Line 340, 3 minat. Please correct.
Please write line 944-976. The authors forgot to write here.
Author Response
Comments and Suggestions for Authors
This is an excellent research paper!! Very important for osteogenesis and angiogenesis research works. However, I have following suggestions:
Please use Italicized texts wherever required.
- 2.1.1. Preparation of PEG: PEG (500 mm) diluted 25 times and final concentration reaching 20 micro Molar. Ease check and correct the unit. Answer:
Answer:
Thanks the valuable comment from the reviewer. We have changed the wording and description in the “Materials and Methods” section “PEG (500 mM) was diluted 25 times and make the final concentration reach to 20 mM” (Page 4, line 158-159)
- Line 231, 234: •100 U/ ml. Please check the •.
Answer:
We have corrected the wording for this mistake. “100 U/ ml” (Page 6, line 249); “100 U/ ml” (Page 6, line 252)
- 2.3.4. Please mention the manufacturer of phalloidin and DAPI, mounting gel.
Answer:
We have included the manufacturer in the “Materials and Methods” section “phalloidin (Sigma, USA)”; “DAPI (Invitrogen, USA)”; and “Gel MountTM (Sigma, USA)” (Page 6, line 288, 289, 292)
- Line 272, 290. At a concentration of 30% to 100%. Please mention percentage of what.
Answer:
We have corrected the wording of “dehydrated at a concentration of 30 % to 100 %” to “dehydrated by ethanol at a concentration of 30 % to 100 %” (Page 6, line 299; Page 7, line 317-318)
- Line 325, 100 ug/m streptomycin. Please make it ml.
Answer:
We thank the valuable comment from the reviewer. We have corrected the mistake to “100 mg/ml” (Page 7, line 353).
- Line 340, 3 minat. Please correct.
Answer:
We have corrected the mistake to “3 min at” (Page 7, line 368)
- Please write line 944-976. The authors forgot to write here.
Answer:
Thanks the kindly remind from reviewer. We have included the description in this section (Page 28, line 988-1002).
Author Contribution: Conceptualization, C.-C.S. and H.-S.H; Data curation, C.-C.S., C.-A.Y., H.-C.C., C.-M.T., Y.-C.Y. and H.-H.H.; Formal analysis, S.-h.H., K.-B.C., C.-A.Y., H.-C.C., C.-M.T. and Y.-C.Y ; Funding acquisition, H.-S.H.; Investigation, S.-h.H. and H.-S.H.; Methodology, C.-C.S. and H.-C.C.; Resources, H.-H.H.; Software, C.-A.Y.; Supervision, H.-S.H.; Validation, C.-C.S.; Visualization, H-S.H.; Writing – original draft, C.-C.S. and K.-B.C.; Writing – review & editing, S.-h.H. All authors have read and agreed to the published version of the manuscript.
Funding: This work was supported by grants from the China Medical University Hospital (DMR-105-057) and the Ministry of Science and Technology (MOST 109-2314-B-075B-011-MY3).
Institutional Review Board Statement: Not applicable.
Data Availability Statement: Data is contained within the article.
Acknowledgments: Murine macrophage Raw 264.7 cells were kindly provided by Professor Hui-Jen Chen (China Medical University, Taiwan).
Conflicts of Interest: The authors have declared that no competing interest exists.

Reviewer 2 Report
In this contribution by Shen and co-workers, the authors prepared gold nanoparticles-modified polyethylene glycol-hydroxyapatite composites for guiding osteogenesis and angiogenesis of mesenchymal stem cells. The results are attractive to the readership of Biomedicines. However, lots of information are missing. The following points should be solved before it could be considered for the publication.
- What’s polydispersity of PEG used in this study?
- This section should not be called as ‘Preparation of Polyethylene Glycol (PEG)’ since PEG is commercially available. It could be changed to ‘Preparation of Polyethylene Glycol (PEG) stock solution’.
- Line 147-149, ‘PEG (500 mM, average molecular weight: 200) (Sigma, USA) was diluted 25 times with deionized water (ddH2O), where 1 ml of PEG was added to 24 ml of ddH2O, with the final concentration reaching 20 μM.’ It is not clear that the final concentration of PEG becomes 20 μM after two rounds of dilution (25*25).
- Line 173-174, what are the thickness of these coating (‘The material used in this experiment was pre- pared by applying the solution to a culture dish and 6 well, 24 well or 96 well plates’)?
- What’s the ligand stabilizing the gold nanoparticles? This may explain the claim in line 416.
- The procedure of how to make the surfaces of PEG, HA, PEG-HA, PEG-HA-AuNP and PEG-AuNP is missing.
- Line 436, ‘The surface morphology of pure PEG had a uniform and homogenous property’. The authors should confirm the molecular weight of PEG used in this study is 200 since PEG200 is in a liquid form. Did the authors perform AFM measurements on a liquid film of PEG200? What’s the thickness of the film?
- In the MTT assay, it is surprising that low molecular weight of PEG doesn’t induce obvious cell toxicity. At the same time, PEG200 should dissolve in the culture medium during incubation.
- What is the water content of the composite material of PEG- HA-AuNP?
- Several recent reviews (doi.org/10.1016/j.actbio.2020.03.043; doi.org/10.1016/j.colsurfb.2020.111462 ) related to this topic should be included.
- Formats issue. Line 23, ‘polyethyl Glycol’ to ‘polyethylene glycol’. Line 93, ‘Poly (Ethylene Glycol)’ to ‘poly(ethylene glycol)’. Line 112, ‘Surface Plasmon Resonance’ to ‘surface plasmon resonance’. Check all.
Author Response
Comments and Suggestions for Authors
In this contribution by Shen and co-workers, the authors prepared gold nanoparticles-modified polyethylene glycol-hydroxyapatite composites for guiding osteogenesis and angiogenesis of mesenchymal stem cells. The results are attractive to the readership of Biomedicines. However, lots of information are missing. The following points should be solved before it could be considered for the publication.
- What’s polydispersity of PEG used in this study?
Answer:
The molecular weight distributions (polydispersity) was 200 kDa used in this study.
- This section should not be called as ‘Preparation of Polyethylene Glycol (PEG)’ since PEG is commercially available. It could be changed to ‘Preparation of Polyethylene Glycol (PEG) stock solution’.
Answer:
Thanks the valuable comment from the reviewer. We have changed the “Preparation of Polyethylene Glycol (PEG)” to “Preparation of Polyethylene Glycol (PEG) stock solution” (Page 4, line 157)
- Line 147-149, ‘PEG (500 mM, average molecular weight: 200) (Sigma, USA) was diluted 25 times with deionized water (ddH2O), where 1 ml of PEG was added to 24 ml of ddH2O, with the final concentration reaching 20 μM.’ It is not clear that the final concentration of PEG becomes 20 μM after two rounds of dilution (25*25).
Answer:
Thanks the valuable comment from the reviewer. We have corrected the wording in the “Materials and Methods” section “PEG 500 μM was diluted 25 times and make the final concentration reach to 20 mM." (Page 4, line 158-159)
- Line 173-174, what are the thickness of these coating (‘The material used in this experiment was pre- pared by applying the solution to a culture dish and 6 well, 24 well or 96 well plates’)?
Answer:
In this study, a fixed amount of liquid was added per unit area of the coating to ensure that the coating has a fixed thickness under different areas (6 well, 24 well or 96 well plates). Since the coating is quite thin, it is not possible to directly measure its thickness. Calculated from the roughness data, the thickness of the composite is between 1-4 nm.
- What’s the ligand stabilizing the gold nanoparticles? This may explain the claim in line 416.
Answer:
Thanks the valuable comment from the reviewer. We have included more detail description in the “Discussion” section and may make it to be more easy follow for this issue. “Focal adhesion kinase (FAK) signaling and actin fiber remodeling were associated with Filopodia formation [65]. Simultaneously, the survival, adhesion and migration of endothelial cells (ECs) also correlate with appropriate organization of cytoskeletal fiber actin and adhesion receptor a5b3 integrin. FAK signaling pathway is the major role for motility of ECs as well as MSCs. Thus, FAK may be the media of various signaling pathways which can mediate the interaction between cells and nanomaterials [66]. Besides, FAK can be activated by integrin clustering, cells bind to ECM (such as fibronectin), or the interaction between growth factors and receptors [67]. Our previous reports had indicated that the average size of AuNPs was approximately at 5 nm [68, 69]. Moreover, the diameter of integrins is about 8 - 10 nm. A literature indicated that the mechanism of AuNPs influence cell migration may be associated with the integrin binding affinity [70]. Thus, the above evidence elucidates the adhesion capacity between AuNPs and cells can be enhanced through a5b3 integrin/FAK signaling pathway which can further facilitates cell migration, proliferation and differentiation capacity in current research.” (Page 25, line 814-827)
- The procedure of how to make the surfaces of PEG, HA, PEG-HA, PEG-HA-AuNP and PEG-AuNP is missing.
Answer:
Thanks the valuable comment from the reviewer. We have addressed more detail description in the “Materials and Methods” section “Different materials (PEG, HA, PEG-HA, PEG-HA-AuNP and PEG-AuNP) were coated on culture dish by applying the materials at the optimal concentration to cover the culture dish, plate or 15 mm round coverslip glass. The coating solution was allowed to adsorb onto the surface of the culture area for 20–30 min. After coating, the residual content of different materials was removed without any washed in order to establish the thin surface coating layer prior to cell culture.” (Page 4, line 189-195)
- Line 436, ‘The surface morphology of pure PEG had a uniform and homogenous property’. The authors should confirm the molecular weight of PEG used in this study is 200 since PEG200 is in a liquid form. Did the authors perform AFM measurements on a liquid film of PEG200? What’s the thickness of the film? Tang
Answer:
The PEG200 (molecular weight =200) is a colorless liquid. Fig 2A was show that AFM measurements on a liquid film of PEG200. The thickness of PEG200 film near 1.5 nm.
- In the MTT assay, it is surprising that low molecular weight of PEG doesn’t induce obvious cell toxicity. At the same time, PEG200 should dissolve in the culture medium during incubation.
Answer:
(1) Thanks for the valuable comment from the reviewer. We have addressed more detailed description in the “Materials and Methods” section “Different materials (PEG, HA, PEG-HA, PEG-HA-AuNP and PEG-AuNP) were coated on culture dish by applying the materials at the optimal concentration to cover the culture dish, plate or 15 mm round coverslip glass. The coating solution was allowed to adsorb onto the surface of the culture area for 20–30 min. After coating, the residual content of different materials was removed without any washed in order to establish the thin surface coating layer prior to cell culture.” (Page 4, line 189-195)
(2) Most PEG200 in the thin layer may still remain adsorbed on the culture surface. One possible reason may be that the culture medium has certain ionic strength which may promote the hydrophobic interaction and binding between PEG and the culture surface. The concentration of PEG dissolved in the culture medium could be too low to cause obvious cytotoxicity.
- What is the water content of the composite material of PEG- HA-AuNP?
Answer:
(1) We don’t understand the mean of “water content” and supposed to be “water contact”.
(2) The mean of The contact angle is a measure of the ability of a liquid to wet the surface of a solid. The hydrophilicity property of materials is critical for the attachment of cells to the Extracellular Matrix (ECM) through cell adhesion molecules. In the present study, the addition of AuNPs into PEG-HA and PEG made the polymer surface more hydrophilic (Figure 1B), suggesting that nanocomposites would facilitate the adhesion ability of MC3T3 and MSCs.
- Several recent reviews (doi.org/10.1016/j.actbio.2020.03.043; doiorg/10.1016/j.colsurfb.2020.111462 ) related to this topic should be included.
Answer:
Thanks the value comment from the reviewer. We have addressed and included the following references suggested by reviewer in the “Introduction” section “Bone homeostasis is maintained by osteoblasts (OBs) and osteoclasts (OCs) within the basic multicellular unit, in a consecutive cycle of resorption and formation. Therefore, a functional scaffold should allow the best possible OB/OC cooperation for bone remodeling, as happens within the bone extracellular matrix in the body [49]. The bone grafts as a current treatment is associated with inherent limitations; hence, the bone tissue engineering as an alternative therapeutic approach has been considered in the recent decades. Concerted participation and combination of the biocompatible materials, osteoprogenitor/stem cells and bioactive factors closely mimic the bone microenvironment. The bioactive factors regulate the cell behavior and they induce the stem cells to osteogenic differentiation by activating specific signaling cascades [50].” (Page 3, line 128-137)
- Formats issue. Line 23, ‘polyethyl Glycol’ to ‘polyethylene glycol’. Line 93, ‘Poly (Ethylene Glycol)’ to ‘poly(ethylene glycol)’. Line 112, ‘Surface Plasmon Resonance’ to ‘surface plasmon resonance’. Check all.
Answer:
We have corrected these mistakes following by reviewer’s suggestion.
(1) “polyethyl Glycol” to “polyethylene glycol” (Page 1, line 23)
(2) “Poly (Ethylene Glycol)” to “poly(ethylene glycol)” (Page 2, line 93)
(3) “Surface Plasmon Resonance” to “surface plasmon resonance” (Page 3, line 112)

Reviewer 3 Report
The paper presents an application of PEG-HA-AuNP nanocomposites. It is a topic of interest to the researchers in the related areas but the paper needs very significant improvement. My detailed comments are as follows:
- The information of MC3T3-E1 cells is not clear, and readers will confuse MC3T3-E1 Subclone 24 cells or MC3T3-E1 Subclone 14 cells.
- MSCs were isolated from the human umbilical cord Wharton’s jelly tissue. However, it lacks MSCs cell identification.
- In Figure 6E and 6F, there is no beta-actin protein and other antibody sources, and the MMP-2 protein bands in the two cells are almost the same.
- In Figure S3B, the pictures for MSC group and PEG group are same.
- The degradability and metabolizability of PEG-HA-AuNP nanocomposites need to be further demonstrated.
Author Response
Comments and Suggestions for Authors
The paper presents an application of PEG-HA-AuNP nanocomposites. It is a topic of interest to the researchers in the related areas but the paper needs very significant improvement. My detailed comments are as follows:
- The information of MC3T3-E1 cells is not clear, and readers will confuse MC3T3-E1 Subclone 24 cells or MC3T3-E1 Subclone 14 cells.
Answer:
Thanks the valuable comment from the reviewer. The MC3T3-E1 Subclone 14 cells line which exhibit high levels of osteoblast differentiation instead of MC3T3-E1 Subclone 24. We have included the more detail description in the “Materials and methods” section “MC3T3-E1 Subclone 14” (Page 5, line 248)
- MSCs were isolated from the human umbilical cord Wharton’s jelly tissue. However, it lacks MSCs cell identification.
Answer:
Thanks for the valuable comment from the reviewer. We have provided the new data in supplemental figure 1 and included the description in the “Materials and Methods” section 2.3.1.” For the characterization of MSCs, the specific surface markers of MSCs were characterized by flow cytometry. In brief, MSCs were detached, washed, and incubated with the indicated antibody conjugated with fluorescein isothiocyanate (FITC) and/or phycoerythrin (PE), against the indicated markers: CD14-FITC, CD34-FITC, CD44-PE, CD45-FITC, CD73-PE, and CD90-PE (BD Pharmingen, USA). PE-conjugated IgG1 and FITC-conjugated IgG1 were used as isotype controls (BD Pharmingen). Next, the antibody conjugated cells were analyzed by FACS analysis. (LSR II, Becton Dickinson, USA). 8th passage of the MSCs were used in this study.” (Page 5-6, line 256-263); in the “Results” section “We also characterized the MSCs phenotypes by detected surface markers of MSCs using a FACS analysis (Figure S1A). The negative surface makers such as CD14, CD34, and CD45, that were expressed in hematopoietic cells, endothelial cells, and immune cells respectively, and the positive surface antigen of MSCs: CD44, CD73, and CD90. Data from FACS analysis were further shown that less than 2% of negative markers (Figure S1B) and higher than 98% of the positive markers were quantified (Figure S1C) and validated respectively in MSCs.” (Page 15, line 540-546); in the “Supplementary Materials” section “Figure S1. Characterization of MSCs using flow cytometry analysis. (A) Cells were harvested and incubated with the respective antibody conjugated with fluorescein isothiocyanate (FITC) and/or phycoerythrin (PE). The indicated markers: CD14-FITC, CD34-FITC, CD45-FITC, CD44-PE, CD73-PE, and CD90-PE (BD Pharmingen, USA). Filled area represents isotype controls. (B) The FACS results of negative matkers, CD14, CD34 and CD45 expression. (C) The FACS data of positive markers, CD44, CD73 and CD90 expression.” (Page 28, line 962-967)
- In Figure 6E and 6F, there is no beta-actin protein and other antibody sources, and the MMP-2 protein bands in the two cells are almost the same.
Answer:
(1) Gelatin zymography is mainly used for the detection of the gelatinases, MMP-2 and MMP-9, respectively. Therefore, b-actin antibody was not used for this assay due to subject to protocol without probed with any primary b-actin antibody .
(2) We have carefully check out and provide the original figure. The MMP-2 expression is really obtained from two different cells, in particular with significantly different on control group (first lane).
- In Figure S3B, the pictures for MSC group and PEG group are same.
Answer:
Thanks for the valuable comment from the reviewer. We have corrected the wrongly representation of Figure S4B (original Figure S3B).
- The degradability and metabolizability of PEG-HA-AuNP nanocomposites need to be further demonstrated.
Answer:
Thanks the valuable comment from the reviewer. We agreed the important issue by reviewer’s suggestion for clinical bone regeneration application. We will take more time for the execution of all these in vivo experiments to elucidate degradability and metabolizability of PEG-HA-AuNP nanocomposites. We hope that the reviewer understand the work is very huge and will be too much to be included in the current paper.

Round 2
Reviewer 2 Report
My previous report focused on the molecular weight of PEG the authors used and the subsequent effect on the other properties. While the authors solved the other questions, they don’t answer the issue of molecular weight properly.
- What’s polydispersity of PEG used in this study?
I was asking the polydispersity of PEG but not molecular weight. However, the author gave the molecular weight of PEG is 200 kDa, but later on they said it was 200 Da. Please confirm the molecular weight of PEG and show the polydispersity.
- Line 173-174, what are the thickness of these coating (‘The material used in this experiment was pre- pared by applying the solution to a culture dish and 6 well, 24 well or 96 well plates’)?
- In the MTT assay, it is surprising that low molecular weight of PEG doesn’t induce obvious cell toxicity. At the same time, PEG200 should dissolve in the culture medium during incubation.
I don’t believe PEG 200 could coat the culture dish without dissolving in the culture medium. Again please confirm the molecular weight.
- What’s the ligand stabilizing the gold nanoparticles? This may explain the claim in line 416.
I was asking about what was the ligand to make the gold nanoparticles. The authors should give the chemical structure of the ligand from Gold NanoTech Inc (Taiwan).
- What is the water content of the composite material of PEG-HA-AuNP?
The composite film was formed on the culture dishes. This film was not totally dried but it wasn’t the solution added in the beginning. It must contain some water. I was asking about the water content of the composite film (PEG-HA-AuNP).
Author Response
My previous report focused on the molecular weight of PEG the authors used and the subsequent effect on the other properties. While the authors solved the other questions, they don’t answer the issue of molecular weight properly.
- What’s polydispersity of PEG used in this study?
I was asking the polydispersity of PEG but not molecular weight. However, the author gave the molecular weight of PEG is 200 kDa, but later on they said it was 200 Da. Please confirm the molecular weight of PEG and show the polydispersity.
Answer:
Thanks for your comments. The molecular weight and polydispersity (PDI) of PEG200 is 200 Da and 1.04, respectively.
- Line 173-174, what are the thickness of these coating (‘The material used in this experiment was pre- pared by applying the solution to a culture dish and 6 well, 24 well or 96 well plates’)?
Answer:
(1) Thanks for your comments. In this study, a fixed amount of PEG solution was added per unit area of the coating to ensure that the coating has a fixed thickness under different areas (6 well, 24 well or 96 well plates). Since the coating is quite thin, it is not possible to directly measure its thickness.
(2) We appreciate the valuable comment from the reviewer. Unfortunately, experimental equipment used for detecting the thickness of these coating was not available in our country now due to COVID-19 pandemic.
(3) However, the film thickness was dependence of PEG molecular weight. In previously study was show that PEG200 film thickness is near to 125 nm. The above result is measured by a profiler (Tensor P-10) and confirmed by FESEM examination of the cross sections.
(4) References:
Shuhui Yu, Kui Yao, Santiranjan Shannigrahi, and Francis Tay Eng Hock. Effects of poly(ethylene glycol) additive molecular weight on the microstructure and properties of sol-gel-derived lead zirconate titanate thin filmsJ. Mater. Res., Vol. 18, No. 3, Mar 2003.
- In the MTT assay, it is surprising that low molecular weight of PEG doesn’t induce obvious cell toxicity. At the same time, PEG200 should dissolve in the culture medium during incubation.
I don’t believe PEG 200 could coat the culture dish without dissolving in the culture medium. Again please confirm the molecular weight.
Answer:
Thanks for your comments. In our study, PEG and PEG composite films were coated on 96 wells plates. The films surface is viscous and not observed that dissolved in culture medium. We believe that a large amount of salt in the medium will inhibit the dissolution of pure PEG, and the formation of intermolecular hydrogen bonds in the composite material strengthens the stability of the membrane structure.
- What’s the ligand stabilizing the gold nanoparticles? This may explain the claim in line 416.
I was asking about what was the ligand to make the gold nanoparticles. The authors should give the chemical structure of the ligand from Gold NanoTech Inc (Taiwan).
Answer:
Thanks for your comments. We have included the more detail description in the “Materials and Methods” section “Gold NanoTech Inc (Taiwan) utilizes unique, and patented technology of physically breaking down gold into nanoparticles, followed by epitaxially stacking these nanoparticles into stacked materials of controlled diameter within the nanometric range. Gold nanoparticle produced by this manufacturing process possesses distinctive physical properties due to a special ionic charge that maintains its structure and is different from commercially available nanogold produced by chemical reduction methods.” (Page 5, line 184-189) Therefore, the surface charge of the gold nanoparticle interacts with the OH functional group on the PEG200 molecular chain then stabilize its dispersion on solution.
- What is the water content of the composite material of PEG-HA-AuNP?
The composite film was formed on the culture dishes. This film was not totally dried but it wasn’t the solution added in the beginning. It must contain some water. I was asking about the water content of the composite film (PEG-HA-AuNP).
Answer:
Thanks for your comments. When composite film was prepared, gold nanoparticles combined with a little water was add into the composite solution. However, water can be completely removed during membrane preparation. Therefore, water content of the composite material of PEG-HA-AuNP "is similar to zero".

Reviewer 3 Report
1) Note that in most cases, PEG-HA-AuNP shows similar biological effects compared with PEG-AuNP. The authors should clarify the advantages after the incorporation of HA.
2) Figure 1A, how to confirm the as-proposed PEG-HA-AuNP structure shown in this scheme?
3) Figure 3A caption, I cannot find any contact angle test in the figure.
4) Figure 3B, the XPS results may only give elemental distribution on the surface of nanocomposites (only a few nm in depth). So what about the case for the whole nanocomposite?
Author Response
- Note that in most cases, PEG-HA-AuNP shows similar biological effects compared with PEG-AuNP. The authors should clarify the advantages after the incorporation of HA.
Answer:
Thanks the new comment from the reviewer. We have addressed and more detail description in the “Introduction” section. “Previous literature demonstrated the effects of cellulose nanofibers (CNF) and hydroxyapatite (HA) nanoparticles incorporating into poly(ε-caprolactone) (PCL) matrix. The surface wettability and mechanical properties of PCL scaffolds were improved by CNF and further with HA addition. The result of compressive and elastic modulus also verified the excellent properties for bone tissue engineering [28]. Furthermore, the addition of HA and CNF did not weaken the biocompatibility of PCL/CNF/HA nanocomposites [28]. Besides, the crystalline HA was proved to be biocompatible, osteoconductive, and exhibited the slowest degradation rate compared with other calcium phosphate [29].” (Page 2, line 95-102)
Reference:
- Morouço, P.; Biscaia, S.; Viana, T.; Franco, M.; Malça, C.; Mateus, A.; Moura, C.; Ferreira, F.C.; Mitchell, G.; Alves, N.M. Fabrication of Poly (-caprolactone) Scaffolds Reinforced with Cellulose Nanofibers, with and without the Addition of Hydroxyapatite Nanoparticles. BioMed research international 2016, 2016.
- Szcześ, A.; Hołysz, L.; Chibowski, E. Synthesis of hydroxyapatite for biomedical applications. Advances in colloid and interface science 2017, 249, 321-330.
- Figure 1A, how to confirm the as-proposed PEG-HA-AuNP structure shown in this scheme?
Answer:
Thanks for your new comments. Gold nanoparticle produced distinctive physical properties due to a special ionic charge that maintains its structure. Therefore, the surface charge of the gold nanoparticle interacts with the OH functional group on the PEG200 molecular chain then stabilize its dispersion on solution. On the other hand, nanoparticle can catch HA particle by molecular force (hydrogen bonding). The above inference is obtained from FTIR data.
- Figure 3A caption, I cannot find any contact angle test in the figure.
Answer:
We have modified the Figure 3A caption to correspond to the figure. “Figure 3. Material characterization. (A) SEM analysis of different materials. (B) The wide scan spectra of PEG and PEG composites by XPS analysis.” (Page 14, line 524-525)
- Figure 3B, the XPS results may only give elemental distribution on the surface of nanocomposites (only a few nm in depth). So what about the case for the whole nanocomposite?
Answer:
Thanks for your new comments. Because this research wants to understand the interaction between surface chemical composition and cells. Therefore, XPS was used to analyze the surface chemical composition. The overall chemical composition can be obtained by ICP-AA. Therefore, we hope that reviewer understand this article does not discuss the influence of the overall composition on the mechanical properties of the films, so it has not been evaluated.

Round 3
Reviewer 2 Report
The ms. is suggested for publication now.
Reviewer 3 Report
I've no further comments now.